# Dual oxidase *Duox* and Toll-like receptor 3 *TLR3* in the Toll pathway suppress zoonotic pathogens through regulating the intestinal bacterial community homeostasis in *Hermetia illucens* L.

Yaqing Huang[1], Yongqiang Yu[1], Shuai Zhan[2], Jeffery K. Tomberlin[3], Dian Huang[1], Minmin Cai[1], Longyu Zheng[1], Ziniu Yu[1], Jibin Zhang[1]*

1 State Key Laboratory of Agricultural Microbiology, National Engineering Research Center of Microbial Pesticides, College of Life Science and Technology, Huazhong Agricultural University, Wuhan, China, 2 Institute of Plant Physiology & Ecology, SIBS, CAS, Shanghai, China, 3 Department of Entomology, Texas A&M University, Texas, United States of America

* zhangjb@mail.hzau.edu.cn

**Data Availability Statement:** All relevant data are within the manuscript and its Supporting Information files. The raw data was upload to NCBI

## Abstract

Black soldier fly (BSF; *Hermetia illucens* L.) larvae can convert fresh pig manure into protein and fat-rich biomass, which can then be used as aquafeed for select species. Currently, BSF is the only approved insect for such purposes in Canada, USA, and the European Union. Pig manure could serve as a feed substrate for BSF; however, it is contaminated with zoonotic pathogens (e.g., *Staphylococcus aureus* and *Salmonella* spp.). Fortunately, BSF larvae inhibit many of these zoonotic pathogens; however, the mechanisms employed are unclear. We employed RNAi, qRT-PCR, and Illumina MiSeq 16S rDNA high-throughput sequencing to examine the interaction between two immune genes (*Duox* in Duox-reactive oxygen species [ROS] immune system and *TLR3* in the Toll signaling pathway) and select pathogens common in pig manure to decipher the mechanisms resulting in pathogen suppression. Results indicate *Bsf Duox-TLR3* RNAi increased bacterial load but decreased relative abundance of *Providencia* and *Dysgonomonas*, which are thought to be commensals in the BSF larval gut. *Bsf Duox-TLR3* RNAi also inactivated the NF-κB signaling pathway, downregulated the expression of antimicrobial peptides, and diminished inhibitory effects on zoonotic pathogen. The resulting dysbiosis stimulated an immune response by activating *BsfDuox* and promoting ROS, which regulated the composition and structure of the gut bacterial community. Thus, *BsfDuox* and *BsfTLR3* are important factors in regulating these key gut microbes, while inhibiting target zoonotic pathogens.

## Introduction

*Hermetia illucens* L. (Diptera: Stratiomyidae) is a saprophytic insect whose larvae (BSFL) consume a wide range of organic wastes and convert them into biomass [1]. BSFL consuming

and SRA, under accession numbers PRJNA600829 and SRP247530.

**Funding:** National Natural Science Foundation of China (31770136); National Key Technology R & D Program of China (2018YFD0500203).

**Competing interests:** The authors declare no conflict of interest.

livestock waste, such as pig manure, inhibit many associated zoonotic pathogen loads. For example, Liu et al. (2008) [2] determined BSFL can reduce *Escherichia coli* in dairy manure. Furthermore, Lalander et al. (2015) [3] discovered that BSFL reduce *Salmonella* spp. as well as viruses in organic wastes.

The mechanisms allowing BSFL to inhibit these zoonotic pathogens have been investigated. Park et al. (2015) [4] characterized an *H.illucens* defensin-like peptide which has activity against Gram-positive bacteria. Elhag et al. (2017) [5] identified seven gene fragments responsible for the production of three types of antimicrobial peptides. And, Zdybicka-Barabas et al. (2017) [6] determined *E. coli*-challenged BSFL had increased phenoloxidase, lysozyme and anti-Gram-positive bacterium activity.

At a much broader scale, insects rely on innate defense reactions to inhibit pathogens by producing antimicrobial peptides, phenoloxidase and $H_2O_2$. In *Drosophila* (Diptera: Drosophilidae), the Toll signaling pathway is mainly induced by Gram-positive bacteria and fungi [7]. In the sea urchin, *Strongylocentrotus intermedius*, petidoglycan and PolyI: C, but not LPS or ZOA, increased the expression of *SiTLR11*, which activated antifungal responses [8]. And, with mosquitoes (Diptera: Culicidae), the intestinal microbiota inhibits the development of *Plasmodium* spp. and other human pathogens through activation of the insect's basal immunity [9].

Toll-like receptors (TLRs) are proteins present in cellular membranes that are capable of recognizing invading foreign bodies (sentinel cells). They are a type I membrane receptor with an extracellular amino terminus and a conserved cytoplasmic region. TLRs recognize specific molecular structures associated with microbial pathogens, which serve to active innate and adaptive immune responses. With routine microbial burdens, such as those found in the absence of infection, the Toll pathway is at low activation levels. However, acute pathogenic bacterial infection transiently increases nuclear factor kappa B (NF-κB)-dependent innate immune signaling.

The insect gut immune system produces microbicidal ROS by dual oxidase (Duox) to restrict the proliferation of invading microorganisms. In addition, ROS is involved in regulating the healing process of intestinal trauma in insects and also functions as a signaling molecule to initiate other self-balancing signaling pathways [10]. The intestinal bacterial community also is associated with host immunity and bacteriostasis. The microbiota modulates anti-pathogen effects of some immune genes plausibly through activating basal immunity [9]. For example, in the oriental fruit fly, *Bactrocera dorsalis*, (Hendel) (Diptera: Tephritidae) ROS, which is induced by the *BdDuox* gene; a gene that plays a key role in intestinal bacterial community homeostasis [11].

ROS serves as an important immune mechanism for many insects against pathogenic microorganisms, such as bacteria, fungi, entomopathogenic viruses, and parasites [12]. For example, when mosquitoes are exposed to *Enterobacter* spp., known to naturally occur in mosquitoes, they are less likely to be infected by *Plasmodium* parasites. ROS activation is suspected to serve as a primary mechanism inhibiting development of the pathogen *in situ* [13].

The Duox regulatory pathway also contributes to maintaining gut–microbe homeostasis in insects [14]. Gut membrane-associated proteins, such as Mesh, regulate *Duox* expression through an arrestin-mediated MAPK/JNK/ERK phosphorylation cascade and play an important role in controlling the proliferation of gut bacteria. Expression of both *Mesh* and *Duox* is correlated with the gut bacterial microbiome, which, in mosquitoes, increases dramatically soon after acquisition of a blood meal [15].

Recent surveys of BSF gut microbiota revealed a diverse community dominated by Bacteroidetes and Proteobacteria [16,17]. The microbiota of the anterior midgut of the BSF contained the greatest microbial diversity, which gradually decreased distally; however, in

contrast, bacterial load increased. The native gut microbiota (i.e., indigenous) of a number of insects has been determined to provide immunity against select pathogens [18,19,20]. For example, the microbial community of the red flour beetle *Tribolium castaneum*, Herbst, (Coleoptera:Tenebrionidae) offers protection against *Bacillus thuringiensis* bv *tenebrionis* [21]. Experiments on silkworm *Bombyxmori*, L., (Lepidoptera: Bombycidae) demonstrated that lactic acid bacteria in the gut enhance host resistance against *Pseudomonas aeruginosa*. Despite the involvement of native gut microbiota in combating infections, the manner by which the immune system of BSFL regulates gut microbiota homeostasis to suppress zoonotic pathogens remains unknown.

In this study, we examined the antimicrobial activity of immune genes dual oxidase (*BsfDuox*) and TLR 3 (*BsfTLR3*) in BSFL, which represent two classic immune pathways. We explored the, (1) reduction in pathogen loads in pig manure by BSFL, (2) expression profile of immune genes *BsfDuox* and *BsfTLR3* in BSFL, (3) expression profile of immune genes after oral pathogenic bacterial challenge, (4) whether suppression of zoonotic pathogens by BSFL is reduced after *BsfDuox-TLR3* RNA interference, and (5) the dynamic change in intestinal bacterial community after RNAi of *Duox-TLR3* genes and their relationship with the suppression of zoonotic pathogens.

## Material and methods

### Rearing and dissection of *H. illucens* L.

The colony of BSF (Wuhan strain) used in this study was located at the State Key Laboratory of Agricultural Microbiology of HZAU. BSFL were reared for 10 days at 27˚C and 70%-80% relative humidity on an artificially sterilized feed (75 g of wheat bran, 75 g of corn flour, and 350 g of water mixed and then autoclaved at 121˚C for 15 min). Third instars were surface-sterilized prior to use. After a 70% ethanol wash for one minute, the BSFL is washed three times in sterile water for one minute. Approximately 60 larvae from each experiment were dissected in sterile distilled water with a sterilized tweezers under a stereo microscope [22]. The dissected guts were transferred to a 2 ml sterile grind tube containing PBS(0.01 M, 1ml) and ground guts into homogenate. The homogenate was used for the different experiments.

### Zoonotic pathogenic bacteria assay in pig manure conversion

One hundred of surface-sterilized BSFL at 8–10 days old were placed into 100 g of fresh pig manure collected from a facility located near the university. In addition, the control group consisted of 100g pig manure. After interfering with the genes of *Duox* and *TLR3*, the transformation experiments were also carried out. $10^5$ CFU/g *Salmonella* spp. and $10^6$ CFU/g *S. aureus*, as well as 100 *Duox-TLR3* RNAi-injected larvae, were added to the 100 g sterilized feed. All treatments were replicated three times. The method for detecting targeted zoonotic pathogens was based on the National Standards of China GB4789.10–2010, in which *Staphylococcus aureus* was detected in 8 days [23]. The methods are described as follows. The selective medium (purchased from Qingdao Hope Bio-technology Co., Ltd) for *S. aureus* was prepared by placing 6.3 g of Baird-Parker agar in 95 ml of distilled water, heated to a boil until completely dissolved, autoclaved at 121˚C for 15 min, and agitated well after sterilization, thereby preventing agar from depositing at the bottom and solidifying. The medium was then cooled to 50˚C, 5 ml of potassium citrate-potassium yolk enrichment solution added, gently agitated, and poured into a plastic plate(diameter:90 mm). To count *S. aureus* colonies, we first collected manure samples (5 g) at different times after larval introduction (0, 2, 4, 6, and 8 days). Samples were mixed with 45 ml of sterile physiological saline in a 100 ml sterile glass conical flask and agitated for 16 min at 180 rpm. The sample was stored at room temperature

(RT) for 5 min prior to the upper layer be sampled. Each sample was diluted using 0.9% saline solution. For each dilution, 0.1 ml was mixed with the appropriate selective medium and plated. Pathogen counts were determined using selective plates, following incubation of plates at 37°C for 48 h. *S. aureus* colonies exhibited a diameter of 2–3 mm, gray to black appearance, light-colored edge, turbid zone, and transparent ring on the outer layer. The mean *S. aureus* count of three plates was expressed as logCFU/g.

Sample processing to perform the counts was performed in the same way for *Salmonella* spp. as for *S. aureus*. Briefly, triplicate plating of the previous samples described was used following GB 4789.4–2016 [24] (Standards Press of China, 2010). The medium used for isolating *Salmonella* spp. growth(Qingdao Hope Bio-technology Co.,Ltd) was prepared by adding 5 g of bismuth sulfite agar to 100 ml of distilled water that was then stirred and boiled for 1 min, cooled to 50°C-55°C, plated, and used the next day. The mean *Salmonella* spp. (CFU/g) of triplicate plates was determined as logCFU/g(Standards Press of China, 2016).

## Total RNA isolation and cDNA synthesis

Total RNA was isolated from black soldier fly at different developmental stages, including eggs, first-fourth instars, adults after mating (13–15 days after eclosion) using RNA extract reagent kit (Invitrogen, Thermofisher, USA).Taking 100 steriled (90% alcohol wash for 1 minute and distilled water wash for 1 minute) eggs in the egg-laying carton of the adult BSF and grind them in trizol extraction kit to extract RNA(Invitrogen, Thermofisher, USA) according to the manufacturer instructions. In addition, total RNA was isolated from different organs and tissues such as gut and the whole body. In addition, total RNA was isolated from RNAi-*Duox-TLR3* and RNAi-*egfp* groups. The experiments were performed three times. RNA quality was analyzed by NanoDrop 2000 spectrophotometry (ThermoFisher Scientific Inc., Waltham, MA, USA) at 260 nm. 1 μg of total RNA was reversely transcribed into first-strand cDNA by using Hiscipt 1st stand cDNA synthesis kit (Nanjing Vazyme Biotechnology Co., Ltd, Nanjing, China).

## Sequence analysis of full-length *BsfDuox* and *BsfTLR3*

The genome of BSF was sequenced and assembled. The genome size of BSF was 1.1 Gb, which is relatively large among the Diptera that has been sequenced [25]. The *BsfDuox* and *BsfTLR3* genes were searched in the BSF genome database by using the Duox protein sequence and TLR3 protein sequence of the fruit fly. The highest similarity sequence of protein was the protein sequence of Duox and TLR3 of BSF. Specific primers for *BsfDuox* and *BsfTLR3* were designed on the basis of the fragment sequence searched from BSF genome database using Premier5.0 software. The transmembrane domains in *BsfDuox* and *BsfTLR3* were identified using TMHMM online software (http://www.cbs.dtu.dk/services/TMHMM-2.0/) and the structural domains of *BsfDuox* and *BsfTLR3* were predicted using the simple modular architectural research tool (SMART, version 7.0, http://smart.embl-heidelberg.de/).

## Microbial oral infection

Third-instar fed an artificial diet supplement with 5% sucrose solution containing concentrated microbe solution ($1 \times 10^8$ colony-forming units (CFUs) per ml).The microorganisms used in this study were zoonotic pathogens, *S. aureus* and *Salmonella* spp. from the State Key Laboratory of Agricultural Microbiology of HZAU. Bacteria used for oral infection were grown in lysogeny broth medium at 28°C and 180 rpm. Exponential microbial culture ($OD_{600}$ = 1.0) was used for all the experiments as previously described [26] and centrifuged for 15 s (8,000 g), then adjusted to the final concentration with aseptic distilled water. The resulting

bacterial counts in each sample were adjusted to approximately $1{\times}10^8$ CFU/ml by aseptic distilled water. We did a preliminary experiment initially to establish appropriate methods. The sample of different dilutions were applied on LB plates after overnight culture at 37˚C, the number of bacteria on the plate was counted to calculate the number of acquired bacteria by per larvae, then measure the turbidimetric OD value of the sample by spectrophotometer, it will take several days. For formal experiment, the bacterial turbidimetric OD value of all the sample was measured by spectrophotometer refer to the turbidimetric OD value of the preliminary experiments results.

Third-instars (8 days old) were fed a 1 ml solution containing concentrated microbes (approximately $1{\times}10^8$ CFU/ml). The larvae fed with a 5% sucrose diet only served as the control. For the analyses of *BsfDuox* and *BsfTLR3* gene expression and ROS level changes after oral infection, the gut samples of different treatments were collected at different times post-oral infection(POI).

## Real-time quantitative PCR (qRT-PCR) analysis

In all cases of gene expression analysis, three independent cohorts of third-instar BSFL were collected for RNA extraction and cDNA synthesis. In the present work, genes, such as *Duox*, *TLR3*, *egfp*, *drosal*, *cecropin*, *ubiquitin*, *dif* and *stomoxyn* were investigated for their expression level at the third-instar stage. The primers for these gene was shown in S1 Table. qRT-PCR was performed using a Bio-Rad CFX system (Bio-Rad, Hercules, CA, USA) with a 384-well plate. Each PCR mixture consisted of 7.8 μl of SYBR Green Mix (Hiscipt 1st stand cDNA synthesis kit, Nanjing Vazyme Biotechnology Co., Ltd, Nanjing, China), 10 nM of each primer, and 2 μl of cDNA (diluted 1:10). The amplification program consisted of pre-incubation at 95˚C for 3 min, followed by 39 cycles of denaturation at 95˚C for 20 s, annealing at 56˚C for 20 s, and extension at 72˚C for 20 s. After the fluorescence quantitative PCR was over, the dissolution curve was analyzed to ensure specific amplification followed by 40 cycles started at 65˚C for 10 s with a 0.5˚C increase for 5 s each cycle until 95˚C. Real-time fluorescence quantitative PCR results were measured by $2^{-\Delta\Delta Ct}$ method as described previously [27]. All the samples were analyzed in triplicate, and the levels of the detected mRNA determined by cycling threshold analysis were normalized using β-actin as the control, The target gene expression is presented as the relative expression levels after normalization. The loads of total bacteria were quantified by qRT-PCR using 16S rRNA gene-specific primers (F 5ʹ−ACTCCTACGGGAGG CAGCAG and R 5ʹ−ATTACCGCGGCTGCTGG) [28] and normalized by using β-actin as the control via a previously described method [29]. PCR mixture consisted of 7.8 μl of SYBR Green Mix (Hiscipt 1st stand cDNA synthesis kit, Nanjing Vazyme Biotechnology Co., Ltd, Nanjing, China), 10 nM of each primer, and 2 μl of cDNA (diluted 1:10). The amplification program was the same as described above. All the samples were analyzed using ANOVA method.

## Measurement of intestinal ROS

The intestine of individual third-instar larvae was hand-dissected in PBS(0.01 M, 1ml). Three independent cohorts of three intestines were used for each measurement. The dissected intestines were ground with PBS solution(0.01 M, 1ml) in the grinder. The quantification of ROS was completed according to the corresponding kit instructions provided by the Institute of Nanjing Jiancheng Bioengineering. In brief, dissected intestines were weighed using a weighing meter, then was added nine times volume of 0.9% saline, centrifuged the sample at 5,000 g for 5 min, the supernatant was further used for the colorimetric quantitative determination of diffused ROS. The absorbance values of each tube were measured under 405 nm by using

UV756 (756 spectrophotometer, Shanghai Opal Instrument Co., Ltd.). The spectrophotometer was preheat for 30 min or more and adjust the wavelength to 405 nm, using distilled water adjust to zero. We detected ROS after a 30-min incubation of the diluted supernatant with the working solution I, II, III, IV(Volume ratio:10:0:1:2:10). Sample was added working solutionI, II, III and centrifugated (4000 g, 25˚C, 10 min) then discarded the supernatant. Adding the working solution IV to solve the precipitation and take 200 μl to the colorimetric dish to determine the absorbance value .ΔOD = determined OD-referenced OD. The protein concentration of each tube was determined by NanoDrop2000 (ThermoFisher Scientific Inc., Waltham, MA, USA). The ROS value of each tube was calculated using the following formula: ROS concentration = (measured OD value−blank OD value) × 163 / (Standard OD value−blank OD value) × sample protein concentration.

## Double-stranded RNA (dsRNA) synthesis and delivery by injection

The *BsfDuox* and *BsfTLR3* sequence fragments were amplified with PCR with premier 5.0 to design of specific primers conjugated with the T7 RNA polymerase promoter. The primer pairs used in dsRNA synthesis are shown in S1 Table.1 μg PCR product was used as the template for dsRNA synthesis utilizing the T7 Ribomax Express RNAi System (Promega, Madison, WI, USA). The dsRNA was isopropanol-precipitated overnight, resuspended in RNase-free $H_2O$, and quantified at 260 nm by using a NanoDrop 2000 spectrophotometer (ThermoFisher Scientific Inc., Waltham, MA, USA) before microinjection. The quality and integrity of dsRNA were determined by agarose gel electrophoresis. The injection condition was set to $P_i$ of 300 hpa and $T_i$ of 0.3 s using an Eppendorf micromanipulation system (Microinjector for cell biology, FemtoJet 5247, Hamburg, Germany). Gene silencing experiments of *Duox-TLR3* RNAi and *egfp* RNAi larvae were performed by injecting 1 μl of 2 μg/μl *dsDuox*-RNA and *dsTLR3*-RNA solution as well as *dsegfp*-RNA into the abdomen of each larva.

## Isolation of bacterial DNA from the gut and high-throughput sequencing

Total bacterial genomic DNA samples from the intestines of 21 individuals were extracted using Fast DNASPIN extraction kits (MP Biomedicals, Santa Ana, CA, USA) according to the manufacturer's instructions and stored at −20˚C prior to further analysis. The quantity and quality of extracted DNA were measured using a NanoDrop ND-1000 spectrophotometer (ThermoFisher Scientific, Waltham, MA, USA) and agarose gel electrophoresis, respectively. PCR amplification of the bacterial 16S rDNA genes to assess the microbial diversity of the BSFL at the V3–V4 regions was performed using 338F (5′ -ACTCCTACGGGAGGCAGCA-3′) and 806R (5′ -GGACTACHVGGGTWTCTAAT-3′). These primers were designed to contain a 7 nt barcode sequence for multiple samples. PCR was performed in a total reaction volume of 25 μl. The PCR conditions were as follows: initial denaturation at 95˚C for 5 min; 30 cycles of 94˚C for 20 s, 58˚C for 30 s, and 72˚C for 30 s; and a final extension at 72˚C for 2.5 min. PCR amplicons were purified with Agencourt AMPure beads (Beckman Coulter, Indianapolis, IN, USA) and quantified using the PicoGreen dsDNA assay kit (Invitrogen, Carlsbad, CA, USA). Purified PCR product libraries were quantified by Qubit dsDNA HS Assay Kit, amplicons were pooled in equal amounts, and pair-end 2 × 300 bp sequencing was performed using the Illumina Miseq PE2500 platform with MiSeq Reagent Kit version 3 (Shanghai Personal Biotechnology Co., Ltd., Shanghai, China). A total of 21 gut samples were subjected to high-throughput sequencing, including three gut samples of 8- to 10-day-old larvae with no treatment of RNAi and 18 gut samples of the *egfp* RNAi- and *Duox-TLR3* RNAi-treated larvae from 4, 8, and 12 days post-RNAi (DPR).

The raw data have been upload to NCBI under BioprojectID PRJNA600829 and SRA under SRP247530. Data analysis was conducted by the bioinformatics software called Quantitative Insights into Microbial Ecology (QIIME, v. 1.8.0) [30]. We get clean sequence reads by removing of the primer sequence, truncation of sequence reads less than an average quality of 20 over a 30 bp sliding window based on the Phred algorithm, and removal of sequences that had a length of <150 bp, as well as sequences that contained mononucleotide repeats of >8 bp [31]. These strict criteria resulted in nearly 94% of the reads being retained.

FLASH (Fast Length Adjustment of Short reads)was used to extend the length of short reads by overlapping paired-end reads for genome assemblies [32]. First, we sorted exactly the same sequence of clean reads according to their abundance and filtered out the singletons and used Usearch tool to cluster under 0.97 similarity. After chimera detection the remaining high-quality sequences were clustered into operational taxonomic units(OTUs) by UCLUST [33]. Selecting the highest abundance sequence from each OTU Library as the representative sequence by using default parameters. OTU taxonomic classification was conducted by BLAST search of the representative sequences set against the 16S database of known species (RDP, http://rdp.cme.msu.edu) [34] using the best hit [35]. Reads that did not match a reference sequence at 97% identity were discarded. Bioinformatics and sequence data analyses were mainly performed using QIIME and R packages (version 3.2.0). Using QIIME software calculates the alpha diversity index of a sample including richness and diversity indices (observed species [Sobs], Chao, abundance-based coverage estimator [ACE], and Shannon) and dissimilarity matrices (Bray–Curtis and weighted UniFrac) [36,37].

## Statistical analysis

Results are shown as the average ± SEM of three independent biological samples. Each experiment was performed three times. Comparison between the two independent samples were performed with student's t-test. Multiple comparisons were conducted by one-way ANOVA (GraphPad Software, La Jolla, CA, USA). Significant level was set at $p < 0.05$. The graphs were also made using GraphPad Prism 5.0 (GraphPad Software, La Jolla, CA, USA).

## Results

### BSFL significantly inhibited pathogenic bacteria in the conversion of pig manure

We investigated the inhibitory action of BSFL in the natural conversion of pig manure. BSFL significantly reduced *S. aureus* and *Salmonella* spp. counts in pig manure from day 2 through day 8. The number of *S. aureus* and *Salmonella* spp. was 5.47 log CFU/g and 6.71 log CFU/g on day zero, respectively. On the eighth day after conversion, the number of *S. aureus* and *Salmonella* spp. reduced to 2.31 log CFU/g and 1 log CFU/g, respectively. BSFL exhibited the greatest success in reducing *S. aureus* from days 2 until days 8. The inhibition rate increased significantly from the fourth to the sixth day, and reached the peak on the eighth day while the number of *S. aureus* in pig manure decreased to the lowest. The inhibition rate of the BSFL to *Salmonella* spp. reached the highest from the second day to the fourth day, after which the inhibition rate dropped to 0 until the eighth day (Fig 1).

### Attenuation of bacteriostasis after *Duox-TLR3* RNAi

After RNAi of *Duox-TLR3* in BSFL, we simulated the experiment of pig manure conversion by BSFL using the sterilized feed. By compared with pig manure, the use of sterilized feed(add only *S.aureus* and *Salmonella* spp.) can reduce the interference of other pathogenic bacteria

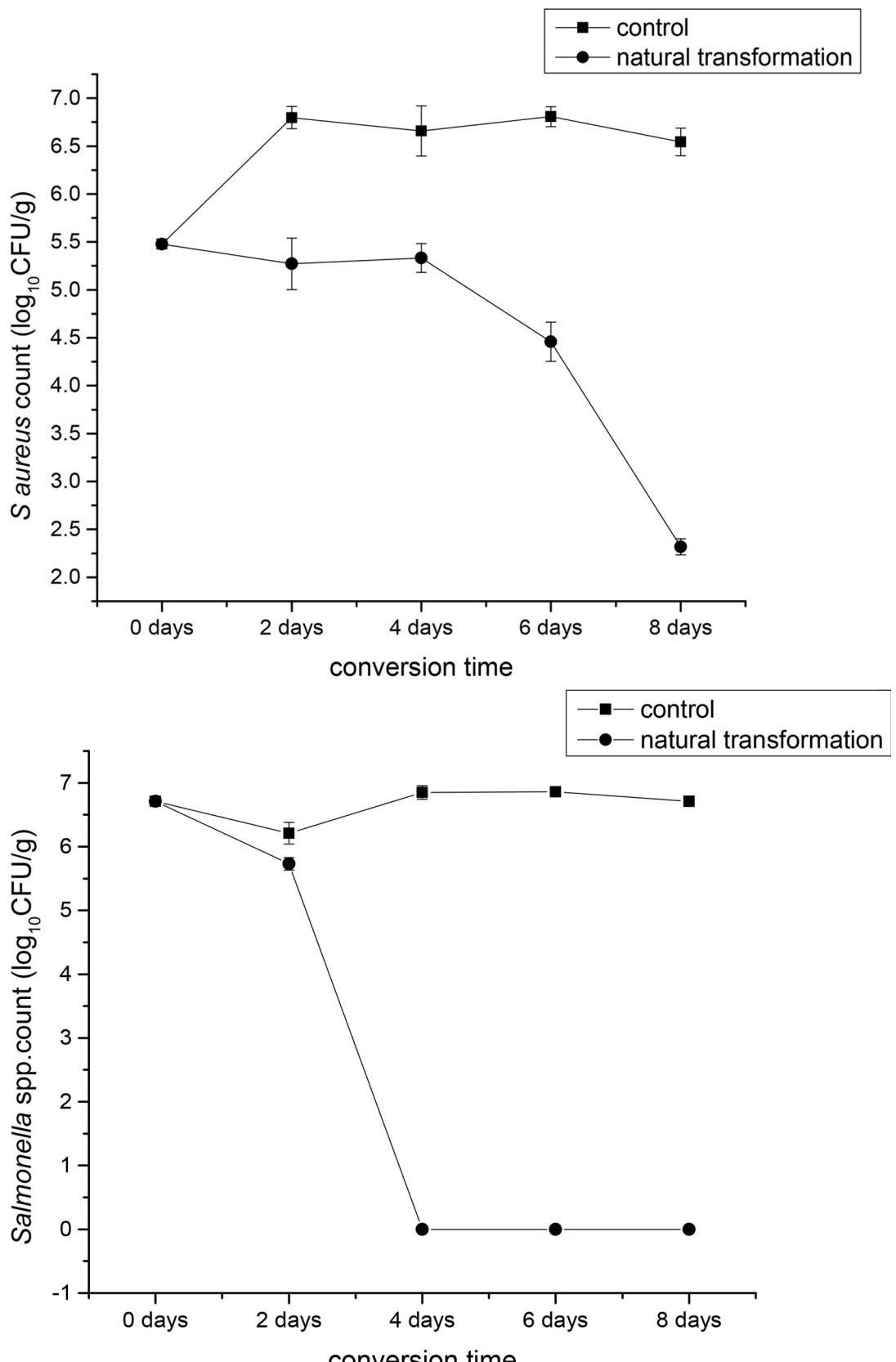

**Fig 1.** Mean *Staphylococcus aureus* (A) and *Salmonella* spp. (B) log CFU/g (mean ± SEM) during 8 days conversion in pig manure with 8-d-old black soldier fly larvae (control) and stored at 27˚C, 60%-70%RH, and a photoperiod of 16:8 (L:D) h in a growth chamber.

(see materials and methods). On day 0 and 8th of conversion, the *S. aureus* count was 6.43 log CFU/g and 6.73 log CFU/g, respectively, and the *Salmonella* spp. count was 5.66 log CFU/g and 5.74 log CFU/g, respectively. In the *Duox-TLR3* RNAi group, the BSFL were also able to inhibit *S.aureus* on day 0 to day 2 of conversion. The inhibitory effect on *S. aureus* was reduced from 2th day to 8th day. Meanwhile, the BSFL were also able to inhibit *Salmonella* spp from day 0 to day 4 of conversion. The inhibitory effect on *Salmonella* spp. was reduced from the 4th to 8th day. Thus, the BSFL induced a diminished effect on pathogen inhibition in *Duox-TLR3* RNAi-injected larvae compared with the natural pig manure transformation conditions (Fig 2).

## Sequence analysis and expression profiles of *BsfDuox* and *BsfTLR3*

We searched the *BsfDuox* and *BsfTLR3* cDNA from the genome database of the BSF. *BsfDuox* was 4.2 kb and encoded 1540 amino acids (S1 Fig), whereas *BsfTLR3* was 1.134 kb and encoded 378 amino acids (S2 Fig). Structural analyses demonstrated that *BsfDuox* consists of FAD-binding and NAD-binding domains and ferric-reduct domain typical of all members of electronic delivery system responsible for $H_2O_2$ generation, as well as extracellular peroxidase homology domain (PHD). The PHD shows myeloperoxidase activity, thereby enabling the conversion of $H_2O_2$ to HOCl in the presence of chloride ion. PHD of Duox is vital for the host immune defense system (S3 Fig). BSFL have one *TLR3* gene, which primes the immune response. Structural analyses demonstrated that *TLR3* had three Leucine-rich repeat domains

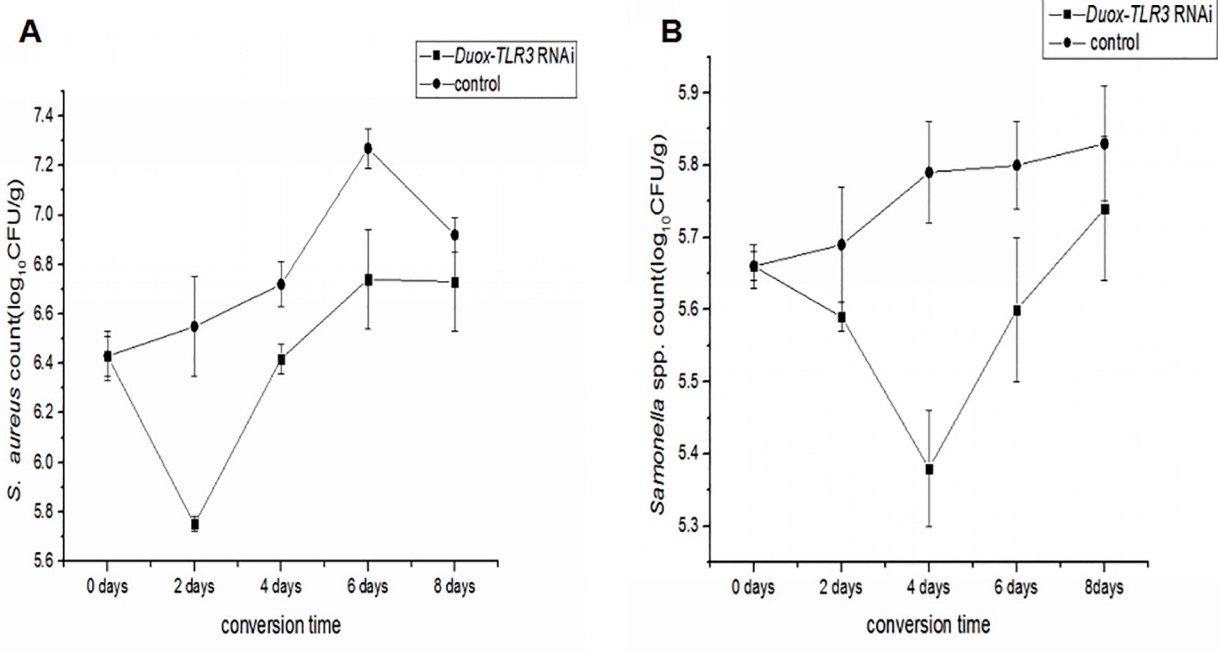

**Fig 2.** Mean *Staphylococcus aureus* (A) and *Salmonella* spp. (B). log CFU/g± SD during 8 days conversion in *Duox-TLR3* RNAi larvae group with (control) 8-d-old black soldier fly larvae and stored at 27˚C, 60–70%RH, and a photoperiod of 16:8 (L:D) h in a growth chamber.

and was responsible for the protein binding and cell signaling functions (S4 Fig). The structure of *BsfDuox* and *BsfTLR3* was similar with the previously reported *Duox* and *TLR3* of *B. dorsalis*, suggesting that they may have the same function.

The *BsfDuox* gene was highly expressed in the egg, first, second, and fourth instar as well as adult stages, but it was weakly expressed in the third-instar larval stage (S5A Fig). The *BsfTLR3* gene was highly expressed in the egg, first, second, third, and fourth instar stages, but it was weakly expressed in the adult stage (S5B Fig). We speculated that the two genes have different expression level to meet their different functions at different stages of BSF development.

## *BsfDuox* and *BsfTLR3* genes were induced upon infection

To investigate the role of *BsfDuox* and *BsfTLR3* in the immune system response, we detected *BsfDuox* and *BsfTLR3* gene expression in the gut upon oral infection with *S. aureus* and *Salmonella* spp. More specifically, *S. aureus* induced a 4.05-times in *BsfDuox* gene expression with a peak at 12 h POI (Fig 3A). Consistent with this, ROS level was increased at 12 h (Fig 3B). *Salmonella* spp. did not activate the *Duox* gene at any point significantly. *Salmonella* spp. and *S. aureus* differently induce the expression of *Duox* gene. Thus, *BsfDuox* gene expression may be regulated by factors secreted by *S. aureus*. To date, uracil secreted by bacteria is the only bacterial ligand [38] that can activate *Duox* activity. Our results showed that *S. aureus* induced a 1.76-fold increase in *BsfTLR3* gene expression at 1 h POI, whereas *Salmonella* spp. induced a 1.75-fold increase in *BsfTLR3* gene expression at 4 h POI. Thus, gram-positive and gram-negative bacteria could induce *TLR3* gene expression (Fig 3C). This study also showed that oral *S. aureus* infection to BSFL could induce the expression of the toll-like receptor pathway nucleic acid transcription factors *Dif* and *Dorsal* to mediate antibacterial activity (Fig 4).

## Sequential difference between the intestinal *Duox-ROS/Toll* signaling pathway after zoonotic pathogen challenge

We studied the temporal differentiation of the immune response of Duox-ROS immunity and Toll signaling pathway by feeding the zoonotic pathogen *S. aureus* to larvae. Infected by zoonotic pathogens could induce the immune response of the Duox-ROS system and Toll signaling pathway in a short time.

*S. aureus* induced a 2.65-fold increase in ROS in the host intestine after 4 h (Fig 3B) but a two-fold increase in *Bsfdorsal* gene in the Toll signaling pathways after 8 h (Fig 4A). The effector gene *Dif* increased by six-fold after 12 h POI compared with the control (Fig 4B). Thus, the immune response of the Duox-ROS system to zoonotic pathogen challenge was earlier than that of the Toll pathway.

## *BsfDuox-TLR3* regulated gut bacterial density

On the basis of the above results (Fig 3), we concluded that the *BsfDuox* and *BsfTLR3* genes were activated differentially with zoonotic pathogens. Immune genes *Duox* and *TLR3* are expressed to control gut microbiota. It was not $H_2O_2$ the only measurable bioactive compound controlling the microbial load. Subsequently, we tested the change in composition of different intestinal symbionts that reacted to the silencing of the *Duox-TLR3* gene by injecting larvae with *ds-Duox-TLR3*. The level of *BsfDuox* gene transcript was inhibited at 4–8 days, *BsfDuox* gene expression varied from 85% to 88% compared with the *egfp* RNAi control (Fig 5A). However, *BsfDuox* expression increased in *Duox-TLR3* RNAi-treated larvae at 10 days and then returned to the basal expression level at 12 days. The level of ROS followed the same pattern with a decrease of 48%-57% as compared with the *egfp* RNAi control at 4–8 days and with an increase of 21% at 10–12 days (Fig 5B). The level of *BsfTLR3* expression was inhibited at 6–10

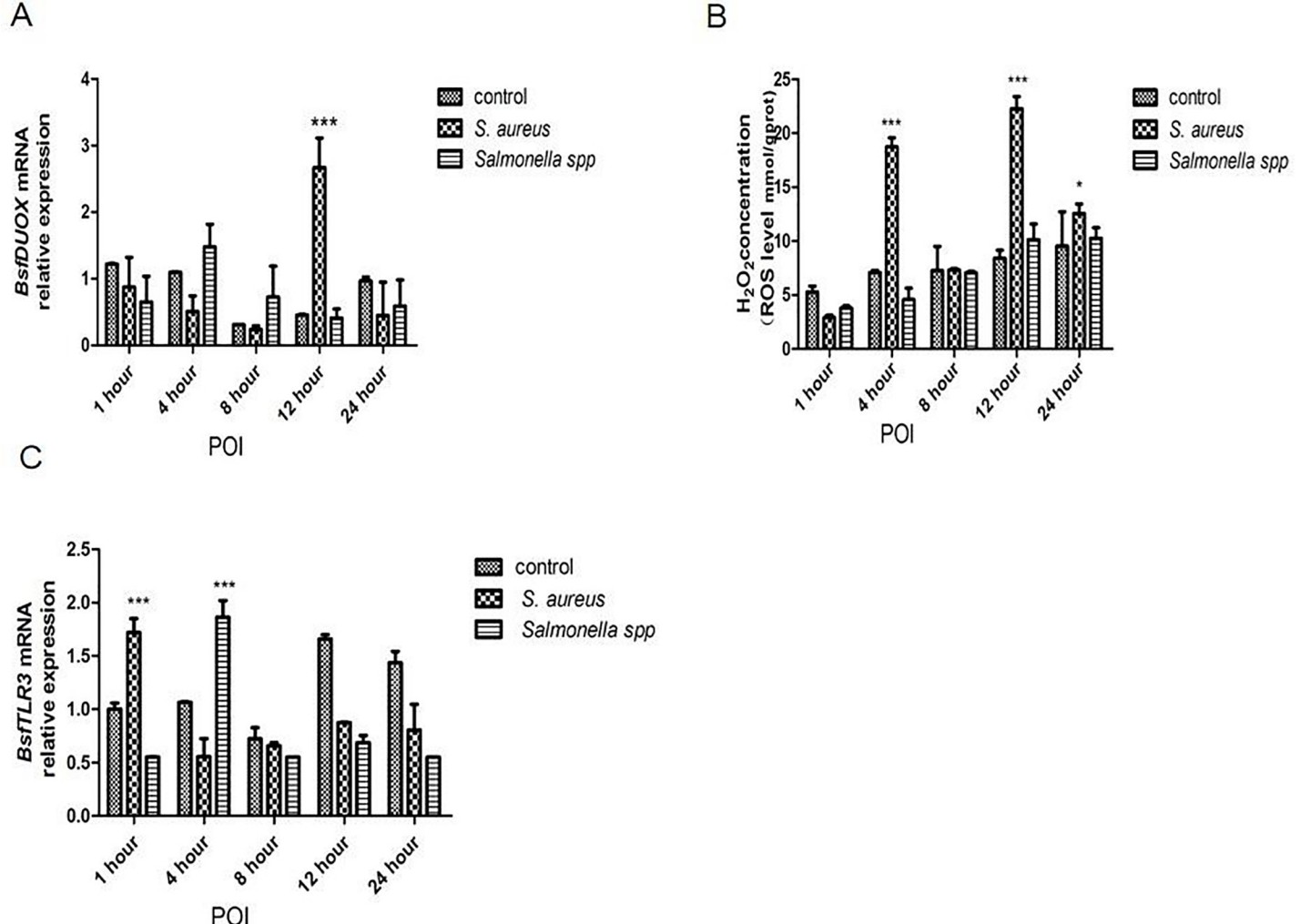

**Fig 3. The response of the *BsfDuox* gene and *BsfTLR3* gene in the gut during oral infection.** (A) Expression levels of *BsfDuox* at different time points in whole guts (without Malpighian tubules) after oral infection with *S. aureus* and *Salmonella* spp. (B) Expression levels of *BsfTLR3* at different time points in whole guts (without Malpighian tubules) after oral infection with *S. aureus* and *Salmonella* spp. (C) The total intestinal ROS levels were quantified with flies at different time points after oral infection. Data are representative of three independent experiments (mean ± SEM). Statistical comparison was based on Student's t–test ($^*p < 0.05$, $^{**}p < 0.01$, $^{***}p < 0.001$). Different letters indicate a significant difference in *BsfD*uox expression and *BsfTLR3* expression and ROS levels among the oral infection with *Salmonella* spp. or *S. aureus*.

days, *BsfTLR3* gene expression varied from 77% to 88% compared with the *egfp* RNAi control. *BsfTLR3* gene expression was 4.48 times higher in *Duox-TLR3* RNAi-treated larvae at 12 days than in the *egfp* RNAi control (Fig 5C). Interestingly, the analysis of the overall bacterial density by qRT-PCR showed that *Duox-TLR3* RNAi larvae had more bacteria at 4–10 days compared with the control. The bacteria load returned to the wild-type level at 12 days (Fig 5D).

BSFL feeding on a diet containing high bacterial loads could induce the production of an expanded spectrum of antimicrobial peptides (AMPs) [39]. However, the expression of AMP was also down-regulated after interfering with two immune genes. The AMP level of cecropin expression in *Duox-TLR3* RNAi larvae decreased by 26.14–36 times compared with that in the control at 8 and 10 days (Fig 6A). The AMP level of ubiquitin expression in *Duox-TLR3* RNAi larvae decreased by 8.71–41.55 times compared with that in the control at 6, 8, and 10 days (Fig 6B). The AMP level of stomoxyn ZH1 expression in *Duox-TLR3* RNAi larvae decreased

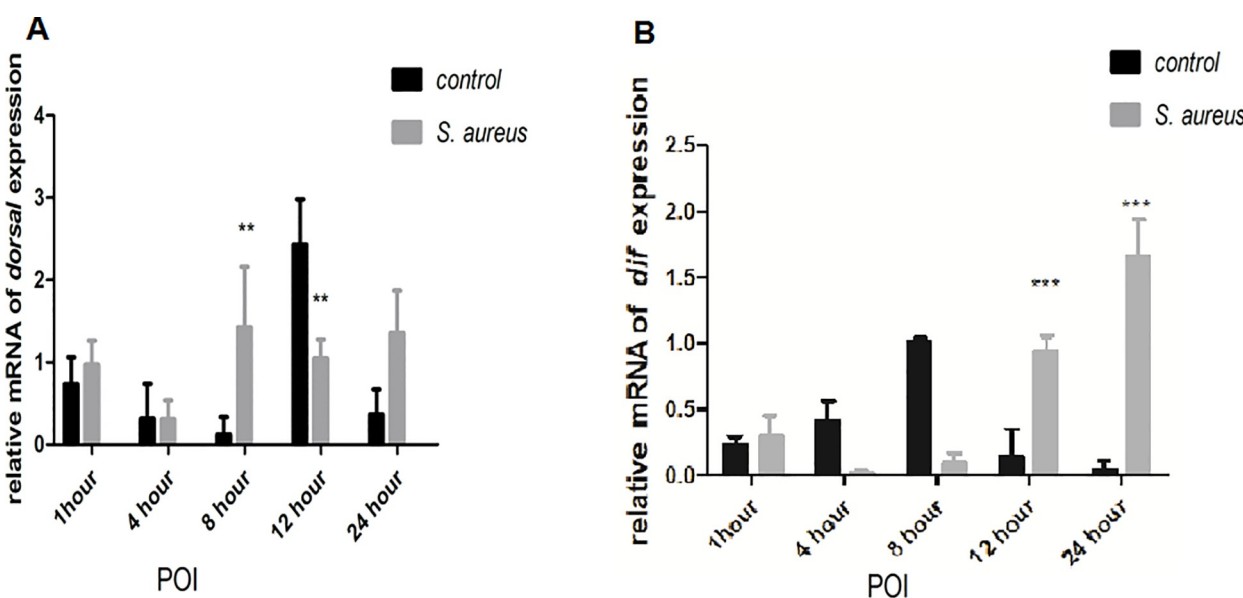

**Fig 4.** (A) The expression level of *Bsf dorsal* gene at different time points after oral infection of *S. aureus* (B) The expression level of *Bsf dif* gene at different time points after oral infection of *S. aureus*. Values are the mean ± SEM of three independent experiments. Statistical comparison was based on Student's t–test(*$p < 0.05$,**$p < 0.01$, ***$p < 0.001$ with Student'S t–test).

by 7.93 times compared with that in the control at 10 days (Fig 6C). The results of 16s DNA sequencing of the intestinal tract of BSF showed that the beneficial symbiont bacteria in *Duox-TLR3* RNAi-injected larvae were 3.4% lower than that in *egfp* RNAi-injected larvae [40] (Fig 7A), whereas the pathogenic bacteria in *Duox-TLR3* RNAi larvae was less than the control group at 4 days while increased by 44.8% compared with that in *egfp* RNAi larvae at 8 days post-RNAi (DPR; Fig 7B). However, these differences were not statistically significant. The depletion of these two immune genes in BSFL reduced AMP and ROS production, leading to the decrease in symbiont bacteria and increase in pathogenic bacteria.

The interference of *BsfDuox-TLR3* gene reduced the ROS and AMP expression levels, resulting in an increase in the number of intestinal bacteria. We attributed the low *BsfDuox* and *BsfTLR3* gene expression levels and ROS levels observed at 4–8 days to the increase in bacteria at a time when RNAi was effective. Low levels of gene expression after interference led to an increase in the total number of bacteria, as well as an increase in the number of pathogenic bacteria, which can stimulate gene expression to return to normal after interference that is the activities that take the observed number of days to recover normal activity. Conversely, high levels of *BsfDuox* gene expression and ROS at 10 days resulted from the decrease in bacteria at a time when RNAi was ineffective. Thus, the larval innate immune genes *BsfDuox* and *BsfTLR3* were in regulating the homeostasis of the gut bacterial and in controlling the bacterial load in the midgut, and exposure to increased pathogens resulted in the increased production of some of these anti-pathogenic factors.

## *BsfDuox-TLR3* regulated the intestinal bacterial community homeostasis

*BsfDuox* and *BsfTLR3* regulated bacterial community density but whether *BsfDuox* and *BsfTLR3* affect the gut bacterial community composition remains unknown. The bacterial composition in *egfp* RNAi-treated and *Duox-TLR3* RNAi-treated larvae was investigated by MiSeq Illumina high-throughput sequencing. The rarefaction curves moved toward saturation

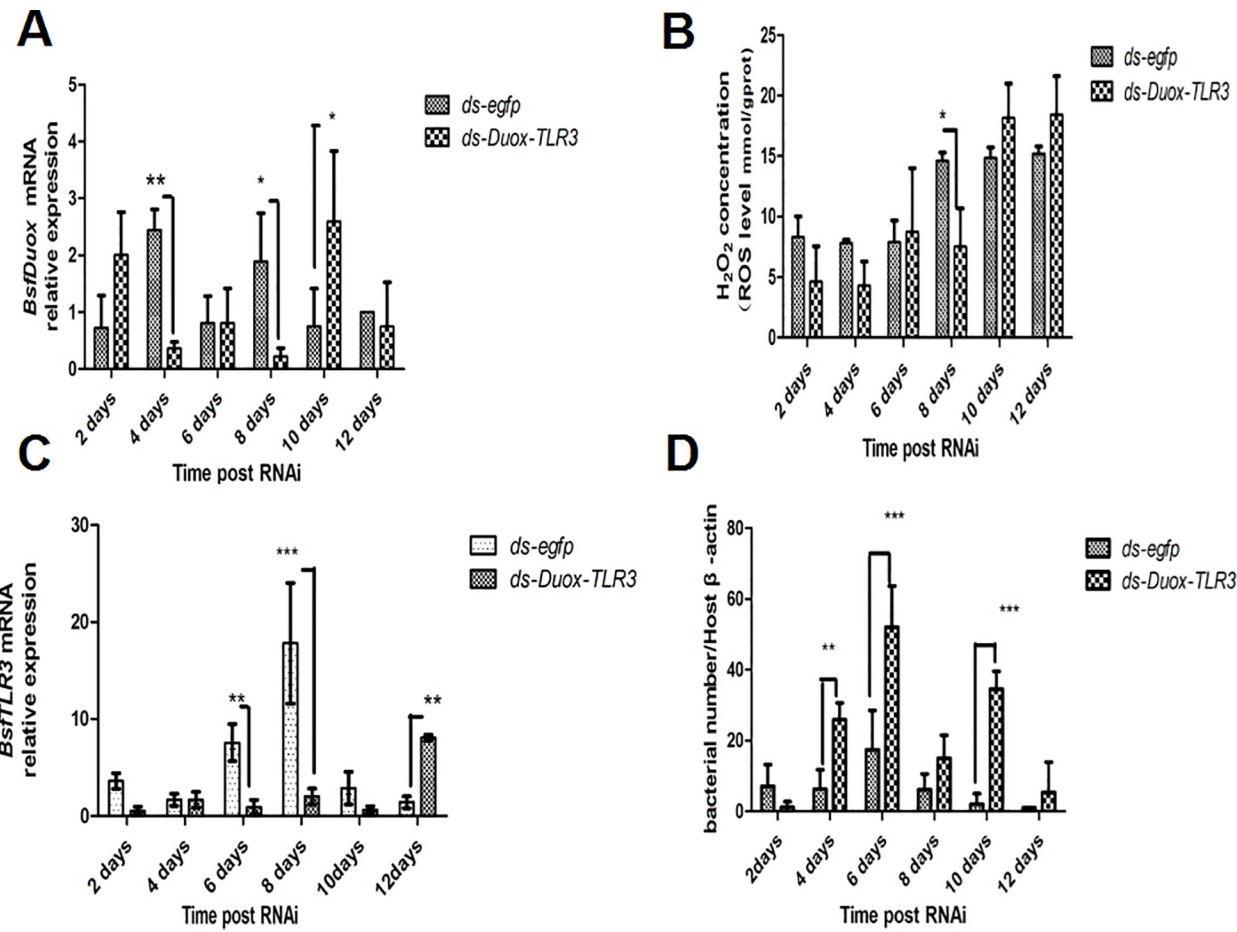

**Fig 5. Interference effects of RNAi of the *BsfDuox-TLR3* gene.** (A) The expression level of the *BsfDuox* gene at different time points after injecting ds-*Duox-TLR3* and ds-*egfp*. (B) The expression level of the Bsf*TLR3* gene at different time points after injecting ds-*Duox-TLR3* and ds-*egfp*. (C) The total intestinal ROS levels at different time points. (D) The total gut bacterial load at different time points. Data are representative of three independent experiments (mean ± SEM). Statistical comparison was based on Student's t–test (*$p < 0.05$, **$p < 0.01$,***$p < 0.001$).

representing the bacterial community well (S6A Fig). Rank abundance curve showed a rich species composition (S6B Fig). Overall, five bacterial phyla were detected in BSF samples, namely, Proteobacteria, Firmicutes, Bacteroidetes, Actinobacteria, and TM7, which composed 63.2%, 18.7%, 16.4%, 1.29%, and 0.41% of the bacterial communities in the gut, respectively. Differences between the bacterial community of *egfp* RNAi and *Duox-TLR3* RNAi samples were calculated using the UniFrac metrics, which measures phylogenetic dissimilarities between microbial communities [41]. Genus- and species-level profiling histograms and principal coordinate analyses based on weighted UniFrac demonstrated great variation in the composition of the gut microbial community upon *Duox-TLR3* RNAi and *egfp* RNAi treatment, especially on days 4 and 8 post-dsRNA injection (Fig 8A). Principal coordinate analysis showed separation of *Duox-TLR3* RNAi- and *egfp* RNAi-treated samples along the major component 1(pc1) and major component 2 (pc2) axes, which explained 44.01% and 21.08% of data variation (S7 Fig), respectively. The analysis of the control *egfp* RNAi samples further confirmed that the intestinal bacterial composition was rich in diversity, and the major genus members of the gut community of BSF were *Providencia*, *Morganella*, *Wohlfahrtiimonas*, and *Dysgonomonas*, whereas the minor members were *Lactococcus*, *Comamonas*, *Pseudomonadaceae*, and *Leucobacter*(Fig 8A).

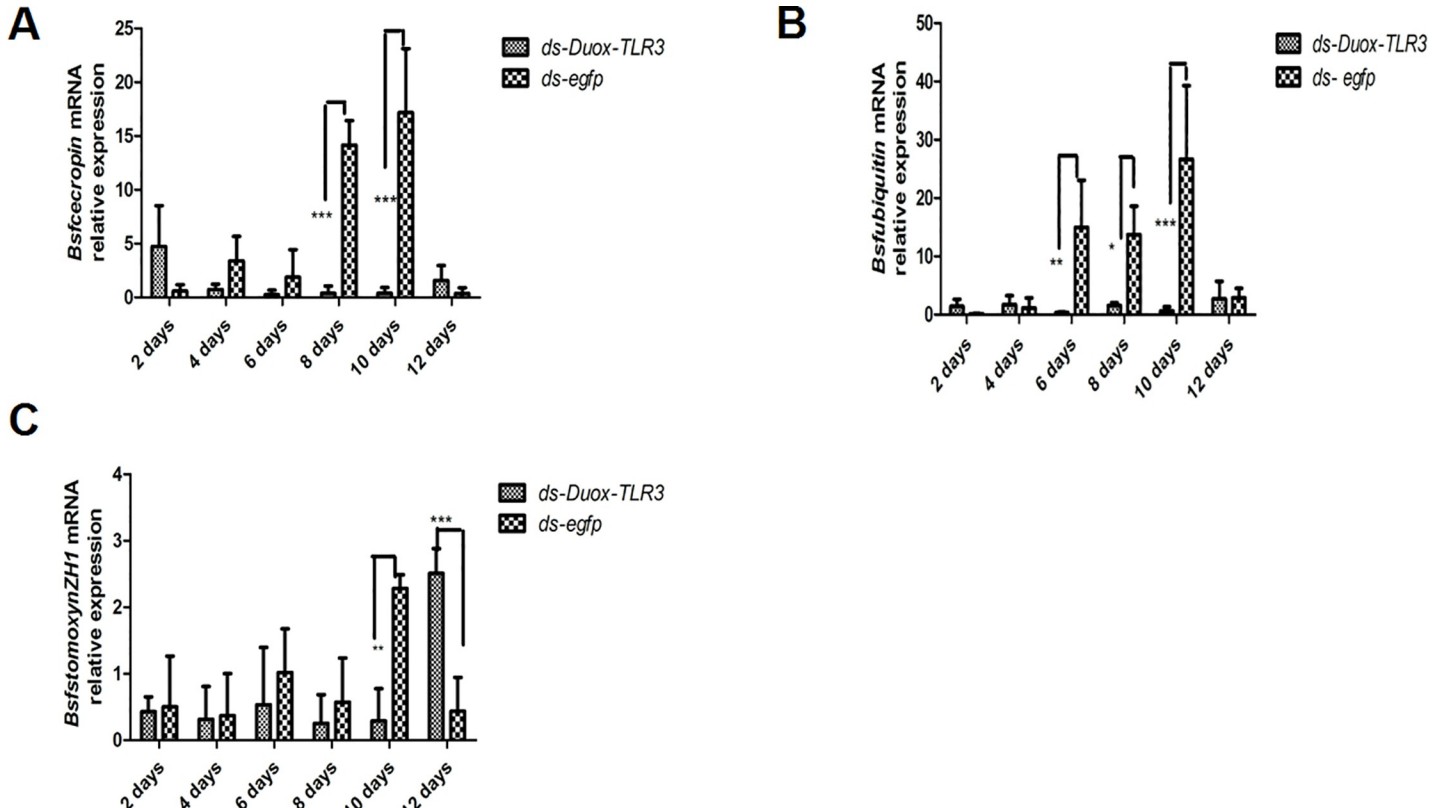

**Fig 6.** The expression level of antimicrobial peptide genes (A) *Bsf cecropin*, (B) *Bsf ubiquitin*, (C) *Bsf stomoxynZH1*, in black soldier fly larvae following the *Duox* and *TLR3* RNA interference. Data are representative of three independent experiments (mean ± SEM). Statistical comparison was based on Student's t–test (*$p < 0.05$, **$p < 0.01$, ***$p < 0.001$).

At 4th day, a decrease of 12% and 14.8% abundance of *Providencia* and *Dysgonomonas* in *Duox-TLR3* RNAi larvae compared with that of control, respectively. However, the relative abundance of others indigenous bacteria increased by 1.17 times in the *Duox-TLR3* RNAi larvae than in the control larvae (Fig 8B). Bacterium *Lactococcus* with low relative abundance increased to detectable levels in the *Duox-TLR3* RNAi larvae compared with that in the control larvae, which in *egfp* group could not reach the minimum level for detection because of their relatively low abundance. The minor group bacterium *Leucobacter* also has certain increase in *Duox-TLR3* RNAi group.(Fig 8C). On the 8th day after dsRNA injection, the relative percentage of *Leucobacter* bacteria in the *Duox-TLR3* RNAi was still significantly higher than that of the control group, The relative proportion of *Lactococcus* bacteria was similar to that of 4th day after dsRNA injection. However, on the 12th day after dsRNA injection, the relative proportion of bacteria in the *Duox-TLR3* RNAi group recovered to the same level as that in the *egfp* control group except to the *Comamonas* bacteria in the *egfp* RNAi group higher than those in the *Duox-TLR3* RNAi group. Thus, the larvae with reduced *BsfDuox* and *BsfTLR3* gene expression displayed a significant difference in bacterial community composition, possibly because of variations in the resistance of bacteria to ROS killing activity. At 8th day, the abundance of *Pseudomonadaceae*, a family of bacteria with low relative abundance in *Duox-TLR3* RNAi larvae increased significantly by two fold compared with that in the controls. The relative abundance of *Comamonas* at 8th day was similar with that at 4th day in *Duox-TLR3* RNAi larvae and was higher than that of

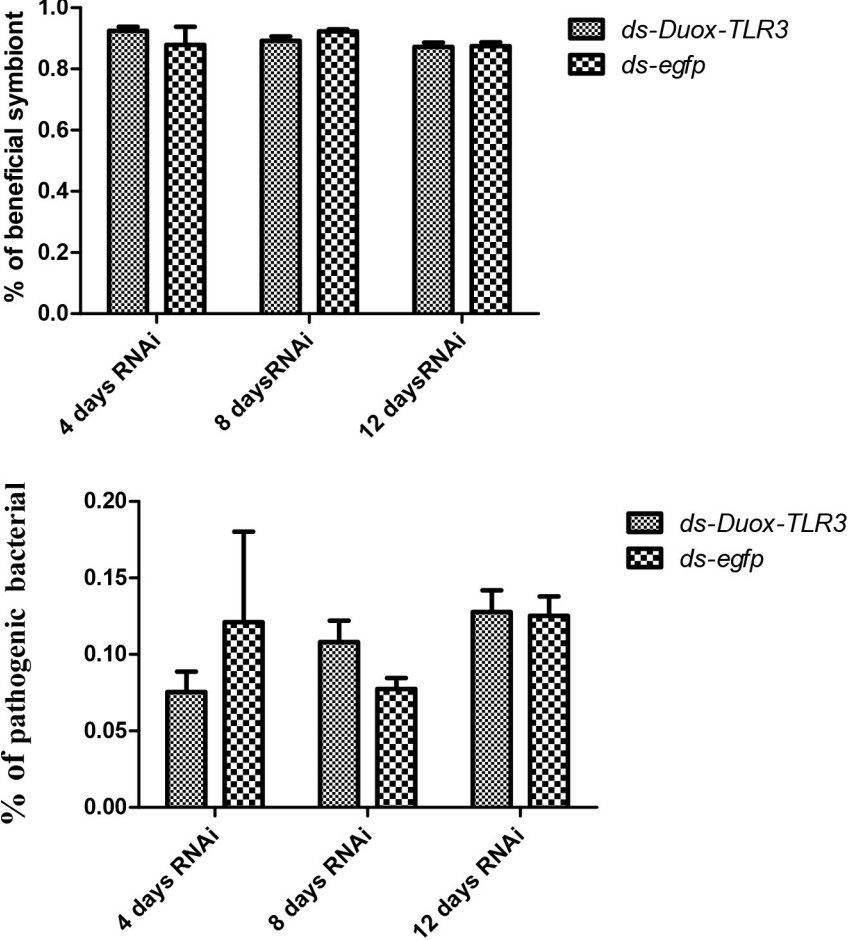

**Fig 7.** (A) The beneficial symbiont in *Duox-TLR3* RNAi larvae and *egfp* RNAi larvae at 4,8,12 days post dsRNA interference (B) The pathogenic bacteria in *Duox-TLR3* RNAi and *egfp* RNAi larvae at 4, 8, 12 days post dsRNA interference. Values are the mean ± SEM of three independent experiments.

*egfp* RNAi group(Fig 8E), which may be due to the reduced ROS and AMP production levels. After *ds-Duox-TLR3* injection at 12[th] day, disordered intestinal bacterial communities stimulated the expression of the *BsfDuox* gene and the production of ROS to suppress non-symbionts. Finally, bacteria taxa composition in *Duox-TLR3* RNAi-treated larvae at 12[th] day return to wildtype, excluding *Dysgonomonas* and *Morganella*, which remained high in *Duox-TLR3* RNAi-treated larvae. Therefore, the *Duox-TLR3* gene had a pivotal role in regulating the structure of the bacterial community in BSFL (Fig 8F). In the present study, the gene interference of *Duox-TLR3* resulted in a decrease in the number of beneficial symbiotic bacteria *Providencia* and *Dysgonomonas* and an increase in the number of conditional pathogenic bacteria *Pseudomonadaceae*, leading to reduced resistance to pathogenic bacteria in the environment.

With the use of PICRUST, different RNAi treatments associated with functional potentials was predicted. The intestinal microbiota of *Duox-TLR3* RNAi-treated larvae was more enriched with genes involved in replication, repair, infectious diseases, carbohydrate metabolism, and amino acid metabolism but less enriched with genes involved in the biosynthesis of other secondary metabolism and enzyme family compared with that of *egfp* RNAi-treated larvae (Fig 9A and 9B).

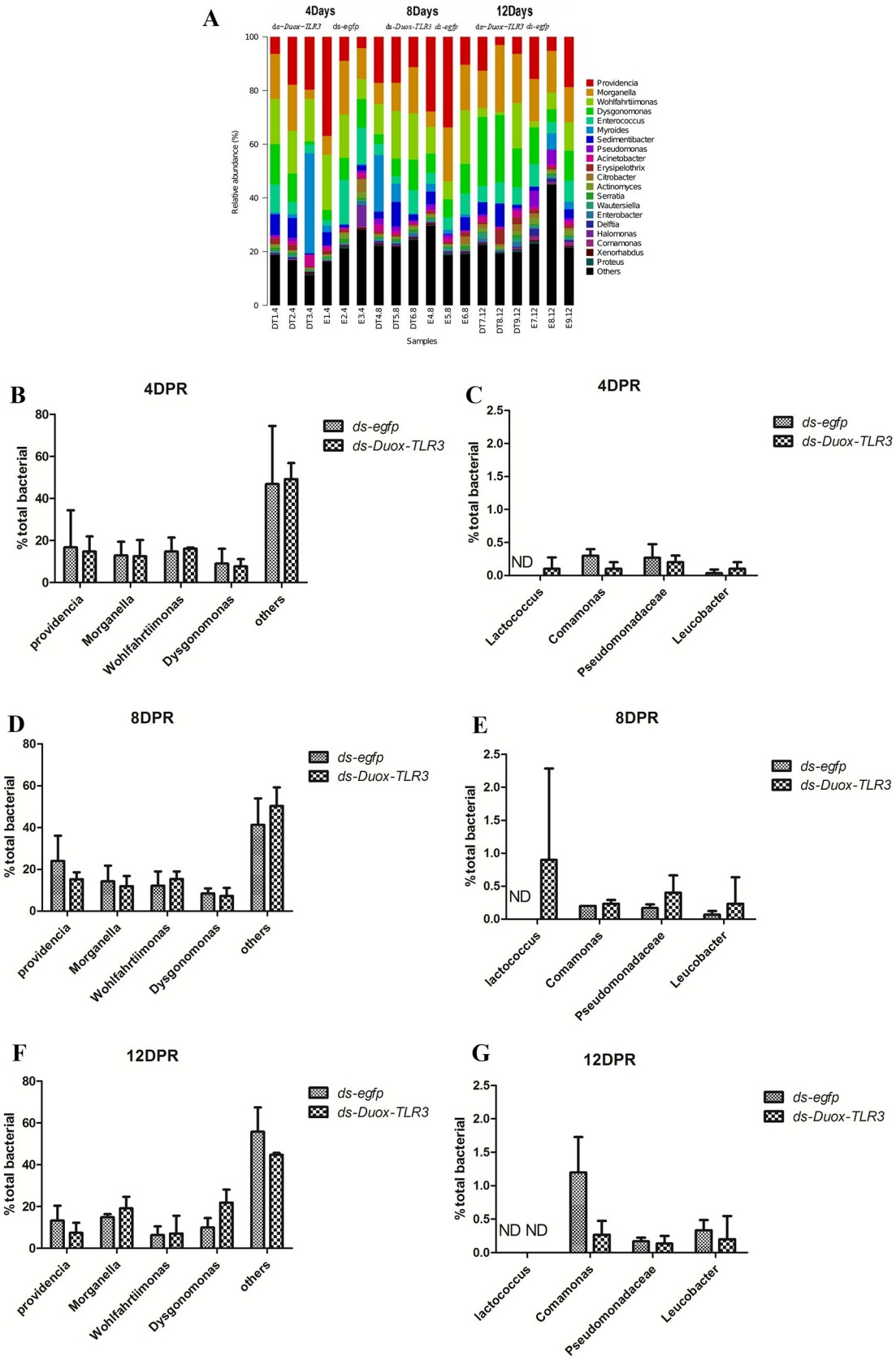

**Fig 8.** *BsfDuox-TLR3* **gene regulates the composition and structure of gut bacterial community.** (A) Taxonomic breakdown at the genus level grouped by ds-*egfp* and ds-*Duox-TLR3* treatments. (B-G) Relative abundance of different bacterial taxa after injecting ds-*Duox-TLR3* and ds-*egfp* at 4, 8 and 12 Day. Data are representative of three independent experiments (mean+s.e.m.). Statistical comparison was based on Student's t–test (*$p$<0.05). ND, not detected.

## Microbial community richness was altered by *BsfDuox-TLR3* RNAi

The effect of *Duox-TLR3* RNAi on microbial diversity was investigated by a standardized approach that evaluated community richness. At 4th day. the Sobs (1192.3 ± 165.7), Chao (1598.9 ± 128.9), ACE(1628.1 ± 145.6), and Shannon(7.1 ± 0.3) indices of the intestinal microorganisms in *Duox-TLR3* RNAi larvae were the same as the Sobs(1351.3 ± 164.7), Chao (1452.8 ± 223.5), ACE(1488.4 ± 202.1), and Shannon(6.8 ± 1.0) indices in the control larvae (Fig 10A). Meanwhile, the richness metrics of Chao and ACE in *Duox-TLR3* RNAi larvae significantly increased at 8th day. In the control larvae, 1387 Sobs were identified, which was almost identical to those in *Duox-TLR3* RNAi larvae. The Chao, ACE, and Shannon indices increased by 31.17%, 31.6%, and 17.4%, respectively, in the *Duox-TLR3* RNAi larvae compared with those in the control larvae at 8th day (Fig 10B).

Finally, no significant difference was noted in the Sobs(1316.0 ± 125.9 vs. 1370.7 ± 151.3), Chao(1464.4 ± 158.8 vs. 1381.3 ± 163.4), ACE(1470.2 ± 142.9 vs. 1398.3 ± 185.1), and Shannon (7.1 ± 0.3 vs. 7.3 ± 0.5) indices between *Duox-TLR3* RNAi and *egfp* RNAi larvae at 12th day(Fig 10C). Thus, *BsfDuox-TLR3* gene silencing by RNAi led to increased bacterial diversity at 8th day compared with the control, possibly because of the low solubility of ROS and AMP.

## Discussion

Intestinal microorganisms directly, or indirectly, affect the immunity of intestinal epithelium cells, and thus impacting the internal environment and development of the host. Microbial populations in the gut regulate host immunity by balancing between an efficient immune response to inhibit foreign pathogens from colonizing and proliferating [42]. Many immune genes of the host are involved in controlling gut microbes. By feeding BSFL zoonotic pathogenic bacteria it was found that the defense system of *Duox-ROS* plays an important role in the intestinal immune defense of BSFL. The silencing of the BSFL *Duox* gene resulted in a significant decrease in the level of ROS in the intestine, indicating that the ROS in the intestine of BSFL is mediated by *Duox*. The Toll signaling pathway plays an important role in the immune response of the BSFL system, zoonotic pathogenic bacteria can rapidly activate *TLR3*, *Dif*, and *Drosal* genes of toll signaling pathways. The zoonotic pathogens that activate *Duox* and *TLR3* to produce ROS and AMPs that act to restrict the growth and proliferation of invading microorganisms [43, 44]. We determined toll pathway-mutant BSFL lacking AMP expression generally express reduced resistance and suffer lethal effects due to gut infection. By contrast, *Duox-TLR3* inactivation leads to uncontrolled increase of bacteria in the gut as demonstrated in this study. That means the immune system relies mainly on two effector molecules, antimicrobial peptides and ROS, to inhibit the colonization of invasive microbes. With the advent of sequencing technology, many bacterial communities in the gut of various insects have been described. Indigenous bacterial community composition was of medium complexity in *B. dorsalis*, which is composed of Porphyromonadaceae, Streptococcaceae, and Sphingobacteriaceae and many unclasssifed bacteria. *BdDuox* silencing led to a decreased abundance of Enterobacteriaceae and Leuconostocaceae in the gut and overgrowth of minor pathobionts [11].

Compared with the intestinal bacteria community of *B. dorsalis*, that of the BSFL was relatively complex. The dominant symbiotic bacteria, *Providencia*, in the gut of BSF belonged to the family of Enterobacteriaceae. Enterobacteriaceae is a commonly found symbiotic taxon in

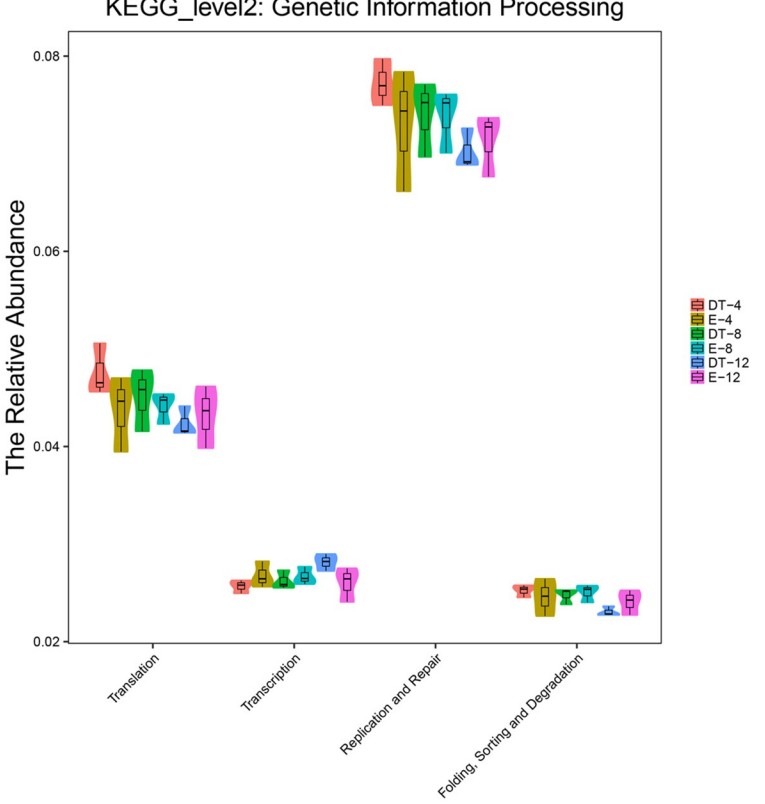

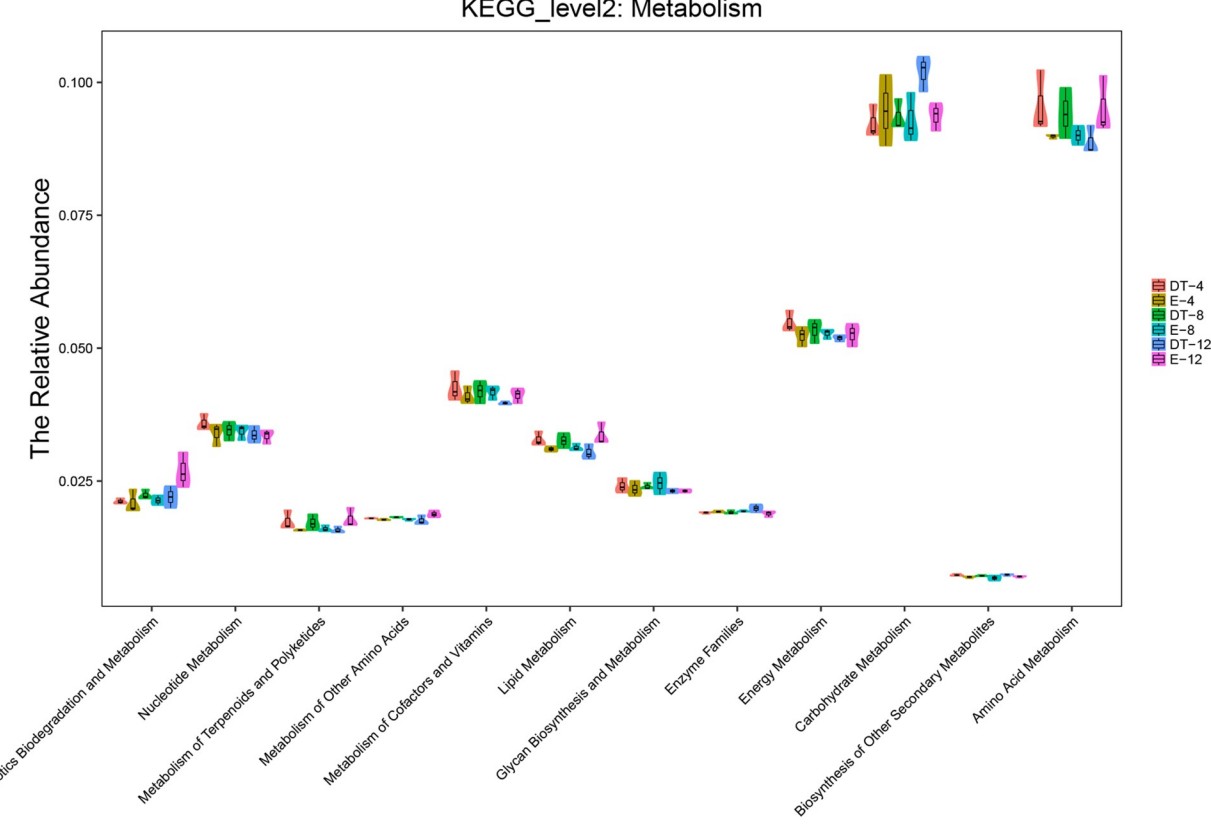

**Fig 9. All of the predicted KEGG metabolic pathways are shown at the second hierarchical level and grouped by major functional categories.**
(A) KEGG_level2: Genetic information processing. (B) KEGG_level2: Matabolism.

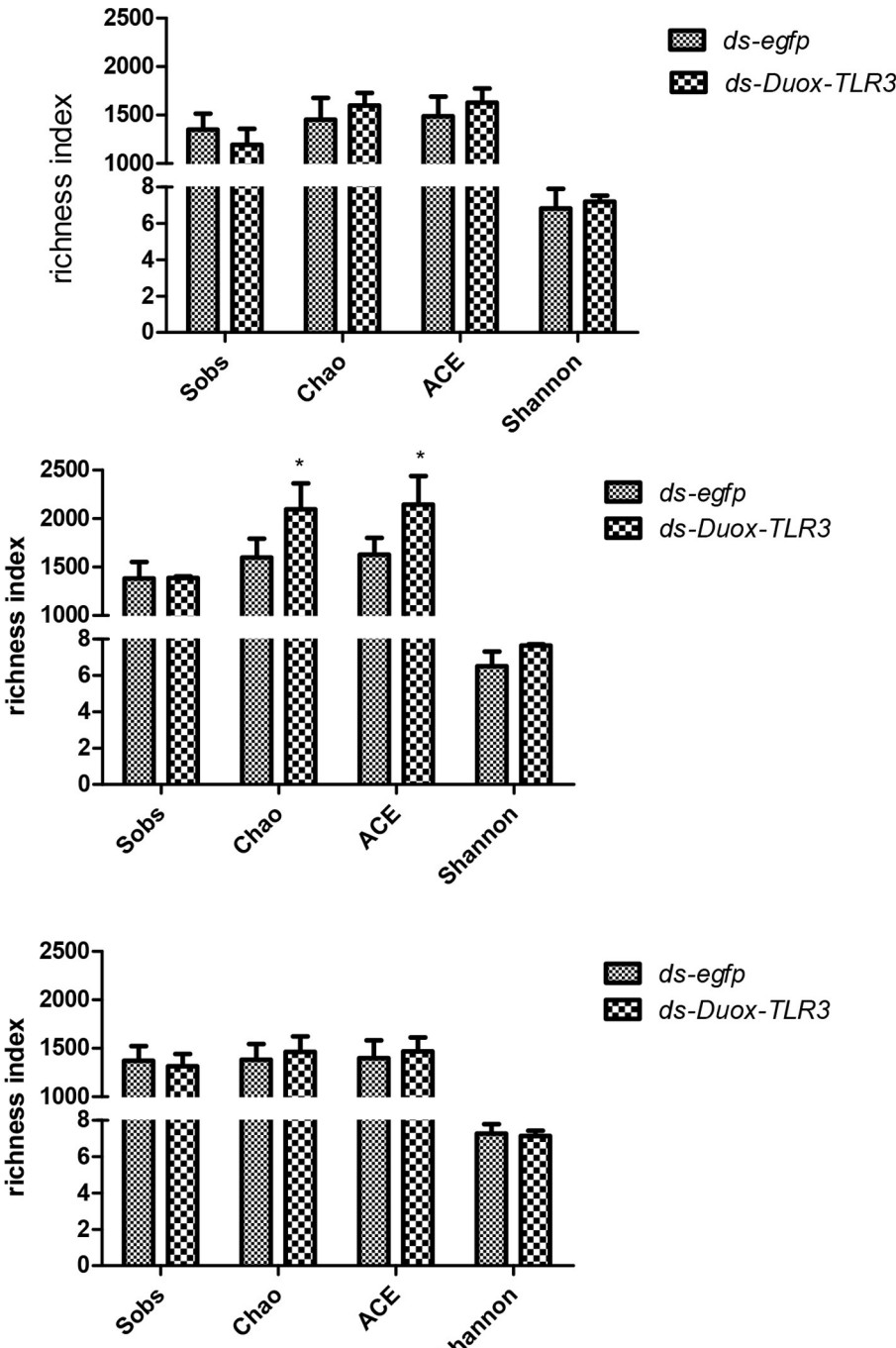

**Fig 10. Effects of RNAi of the *BsfDuox-TLR3* gene on diversity metrics.** Richness measured as observed operational taxonomic units (OTUs; Sobs), Chao, ACE and Shannon indices of gut bacterial communities from different treatment at three timepoints. (A) 4 DPR. (B) 8 DPR. (C) 12 DPR. Data are representative of three independent experiments (mean ± SEM). Statistical comparison was based on Student's t–test ($p$).

insects, and specifically belonging to the γ-proteobacteria, a class that includes dominant symbiotic bacteria in many insect lineages [45]. Enterobacteriaceae, as dominant commensal bacteria in the intestinal tract of insects, may play an important role in improving host fitness by preventing the colonization of foreign pathogens [46] and contribute to nitrogen fixation [47]. *Providencia* spp. are gram-negative bacteria that belonged to the Enterobacteriaceae. *Providencia* spp. resist two kinds of antibiotics including colistin and tigecycline, making *Providencia* multidrug-resistant. *Providencia* spp. resistant to carbapenem antibiotics are increasingly reported. Meanwhile, *Dysgonomonas*, which belong to coccobacilli is a facultative anaerobic bacteria [48]. *Dysgonomonas* spp. is a major group represented in the gut microflora of BSFL, and *Bactrocera tau*. *Dysgonomonas* spp. isolated from human blood samples show antimicrobial susceptibility that directly inhibits competitors and potent pathogens from the same niche [49].

In this study, BSFL were determined to use the Duox-ROS immune system and Toll signaling pathway as a means of intestinal immune defense. Gene expression profiles of *BsfDuox* and *BsfTLR3* at different development times indicated that they may have key functions in host development. The role in immune defense of *Duox* and *TLR3* has been studied well, but their combined role in the regulation of intestinal microbial homeostasis and bacteriostasis is rarely reported. In this study, the silencing of the target gene *BsfDuox-TLR3* was successfully achieved for 4 days. The expression level of the *BsfDuox* was downregulated at 4–8 days of treatment, whereas that of the *BsfTLR3* was downregulated at 6–10 days of treatment. The maintenance of the RNAi effect in a short time is ideal for the study of changes in intestinal microflora homeostasis. Furthermore, silencing of the *BsfDuox-TLR3* gene in larvae led to the increase in density and changes in composition and diversity of intestinal indigenous bacterial community. This finding was similar with the report on the inactivation of *Duox* and intestinal bacterial reproduction in *B. dorsalis* [11].

At the 10-12[th] day after dsRNA injection, the disordered intestinal bacterial community in turn stimulated BSFL *Duox-TLR3* gene expression and ROS production to suppress the overgrowth of non-symbiotic bacteria. Finally, at the 12[th] day after dsRNA injection, BSFL*Duox-TLR3* gene regulated the structure and composition of the host intestinal bacterial community to a normal level. The relative proportion of dominant intestinal commensal bacteria *Dysgonomonas* and *Providencia* decreased, whereas that of *Lactococcus* and *Pseudomonadaceae* increased in *Duox-TLR3* RNAi-treated larvae at 4–8 days. *Dysgonomonas* and *Providencia* in the gut of BSF may enhance inhibition to *Salmonella* spp. and *S. aureus*, a significant decrease in *Dysgonomonas* and *Providencia* by *Duox-TLR3* silencing led to an adverse effect on the host [50]. Meanwhile, *Pseudomonadaceae* is a minor component in the BSF gut and includes four pathogenic bacteria, namely, *P. aeruginosa* [51], *Pseudomonas pertucinogena* [52], *Pseudomonas putida* [53], and *Pseudomonas fluorescens* [54]. *P. aeruginosa* is a human pathogen associated with human skin, mucous membrane, intestinal tract, and upper respiratory tract. The increase in *Pseudomonadaceae* caused by silencing of the *BsfDuox-TLR3* could cause toxicity to the host. Thus, silencing the *BsfDuox-TLR3* gene in the intestines allowed the increase of harmful bacteria.

The result may suggest that the depletion of these two immune genes in BSFL resulted in a proliferation of the minor pathogenic bacteria, which may have proven the potential decrease in anti-environmental pathogenic bacteria immune responses.Therefore, *BsfDuox* and *BsfTLR3* could regulate homeostasis by ensuring the stability of symbiotic bacteria and suppressing excessive growth of minor pathogen bacteria. Gut microbiome disorders are reminiscent of inflammatory bowel disease in humans, but intestinal symbiotes can also cause bowel disease under certain circumstance [55,56]. In humans, many intestinal mucosal inflammatory diseases arise from abnormal intestinal and microbial relationships [57]. Therefore, the study on the role of BSFL *Duox-TLR3* gene in the homeostasis of intestinal bacterial community will

provide reference for the further exploration of the maintenance mechanism of healthy intestinal microorganisms.

Overall, similar to humans, BSF larval intestine harbors a natural microbiota, participating in host metabolism and provide nutrients [58] as well as in the degradation of harmful substances [59] to protect the host from adverse factors such as natural enemies, parasites. In this study, a representative of the predominant gut immune gene *Duox-TLR3* from BSF showed antimicrobial activity that directly prevented the emergence and overgrowth of minor pathobionts. The mode of protection against an encountered pathogen was possibly due to the persistent immune responses involving free radicals and antibacterial peptides. By interfering with the *Duox-TLR3* gene, the intestinal bacterial richness and composition changed (*P. putida*, and *P. fluorescens*) to provoke chronic inflammation under dysbiosis conditions and weakened suppression on *S. aureus* and *Salmonella* spp.. Therefore, the natural bacterial flora is crucial to maintain the stable balance of intestinal microbial communities, resisting and eliminating foreign pathogenic bacteria for insect physiological ecology such as growth and development. Our results demonstrated that *BsfDuox* and *BsfTLR3* could regulate the gut key bacteria *Providencia* and *Dysgonomonas* homeostasis to depress zoonotic pathogens.

## Supporting information

**S1 Fig. The nucleotide and deduced amino acid sequences of BsfDuox cDNA of black soldier fly.**
(PDF)

**S2 Fig. The nucleotide and deduced amino acid sequences of BsfTLR3 cDNA of black soldier fly.**
(TIF)

**S3 Fig. The domain structures of the *BsfDuox* in black soldier fly.**
(TIF)

**S4 Fig. The domain structures of the *BsfTLR3* in black soldier fly.**
(TIF)

**S5 Fig. Expression profiles of the *BsfDuox and BsfTLR3* at different development stages.**
(TIF)

**S6 Fig. Rarefaction curve and rank-abundance curve based on bacterial OTUs at dissimilarity level of 3%.** (A) Rarefaction curve. (B) Rank-abundance curves.
(TIF)

**S7 Fig. Principal coordinate analysis based on weighted UniFrac metrics.** The result showed a separation of *ds-Duox-TLR3* and *ds-egfp*-treated samples along the first two axes, which explained 44.01% and 21.08% of the data variation, respectively. (A) 3Dscore plot of the *ds-Duox-TLR3* RNAi and *ds-egfp* RNAi sample (B) 2D biplots on PC1–PC2 plane overlapping scores and loadings.
(TIF)

**S1 Table. Primers used in this study.**
(DOCX)

## Author Contributions

**Data curation:** Yaqing Huang, Shuai Zhan, Dian Huang.

**Investigation:** Yaqing Huang, Yongqiang Yu, Dian Huang.

**Methodology:** Yaqing Huang, Yongqiang Yu.

**Software:** Shuai Zhan.

**Supervision:** Jibin Zhang.

**Writing – original draft:** Yaqing Huang.

**Writing – review & editing:** Jeffery K. Tomberlin, Minmin Cai, Longyu Zheng, Ziniu Yu, Jibin Zhang.

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
