## [Decision Letter · Decision Letter 0]

30 Dec 2019

PONE-D-19-31548

Dual oxidase gene Duox and Toll-like receptor 3 gene TLR3 in the Toll pathway suppress zoonotic pathogens through regulating the intestinal bacterial community homeostasis in Hermetia illucens L.

PLOS ONE

Dear Dr. Zhang,

Your manuscript has been reviewed by four experts in the field and their comments follow. As detailed in their critiques, all reviewers made specific suggestions that will substantially improve the presentation and technical quality of your work.

After careful consideration, we feel that your paper has merit but does not fully meet PLOS ONE’s publication criteria as it currently stands. Therefore, we invite you to submit a revised version of the manuscript that addresses the points raised during the review process.

We would appreciate receiving your revised manuscript by Feb 13 2020 11:59PM. To enhance the reproducibility of your results, we recommend that if applicable you deposit your laboratory protocols in protocols.io, where a protocol can be assigned its own identifier (DOI) such that it can be cited independently in the future. For instructions see: http://journals.plos.org/plosone/s/submission-guidelines#loc-laboratory-protocols

We look forward to receiving your revised manuscript.

Kind regards,

Dong-Yan Jin

Academic Editor

PLOS ONE

Additional Editor Comments:

Revised paper will be re-reviewed by the original four reviewers if they are available.

Journal Requirements:

2.  We note that you are reporting an analysis of a microarray, next-generation sequencing, or deep sequencing data set. PLOS requires that authors comply with field-specific standards for preparation, recording, and deposition of data in repositories appropriate to their field. Please upload these data to a stable, public repository (such as ArrayExpress, Gene Expression Omnibus (GEO), DNA Data Bank of Japan (DDBJ), NCBI GenBank, NCBI Sequence Read Archive, or EMBL Nucleotide Sequence Database (ENA)). In your revised cover letter, please provide the relevant accession numbers that may be used to access these data. For a full list of recommended repositories, see http://journals.plos.org/plosone/s/data-availability#loc-omics or http://journals.plos.org/plosone/s/data-availability#loc-sequencing.

Reviewers' comments:

Reviewer's Responses to Questions

**Comments to the Author**

1. Is the manuscript technically sound, and do the data support the conclusions?

Reviewer #1: Yes

Reviewer #2: Partly

Reviewer #3: Partly

Reviewer #4: Yes

2. Has the statistical analysis been performed appropriately and rigorously? 

Reviewer #1: Yes

Reviewer #2: No

Reviewer #3: I Don't Know

Reviewer #4: Yes

3. Have the authors made all data underlying the findings in their manuscript fully available?

Reviewer #1: Yes

Reviewer #2: Yes

Reviewer #3: Yes

Reviewer #4: No

4. Is the manuscript presented in an intelligible fashion and written in standard English?

Reviewer #1: No

Reviewer #2: No

Reviewer #3: Yes

Reviewer #4: Yes

5. Review Comments to the Author

Reviewer #1: This study focused on the mechanism black soldier fly larvae used to inhibit two pathogens, S. aureus and Salmonella. To do so, two immune genes were silenced, upon which two symbionts were observed to decrease in abundance and inhibiting effects on the pathogens were reduced. This demonstrated that the immune genes under study play a role in the pathogen reducing effect of native BSFL.

The paper brings important and new information and is highly relevant for the application of BSFL as waste converter, since waste streams can contain the pathogens under study. However, the language use was very poor and disappointing, and it reduced the readability of the paper. The many language errors deduct the reader from the content and devaluate the manuscript. As a reviewer, I corrected a number of language errors, but certainly not all of them. I strongly recommend professional language editing. On some places in the paper, statements were formulated too general and conclusions were drawn too strong in my opinion, i.e. based on evidence that was not substantial enough to make that conclusion. Also some elements are missing in the materials and methods section. I believe this manuscript can become a very meaningful paper as it contains interesting results but it needs major revisions.

Lines 17-19: “larvae can convert fresh pig manure into protein and fat-rich biomass, which can then be used as livestock feed Currently, 19 it is the only insect approved for such purposes in Europe”. Please note that Hermetia is currently allowed in Europe for use in fish feed only, and not in feed for other animal species. In addition, even when used in aquafeed, the insect is not allowed to be reared on manure. Hence, lines17-19 are confusing and appear to say that much more is legally allowed in Europe than is currently the case. Please rephrase.

Line 21 “BSF larvae inhibit these zoonotic pathogens”. Here to, the claim is too general: there are some indications in literature that show that larvae can inhibit growth of these pathogens, but these were only a number of occasions and this is certainly not yet demonstrated on all substrates and in all rearing conditions. A claim such as “BSF larvae inhibit these zoonotic pathogens” might give rearers and other stakeholders the idea that these pathogens, whenever present in the substrate will never be a problem. However, correct communication in sciences is needed to avoid misunderstandings and wrong conclusions in the insect sector. Please rephrase.

Line 28: “Providenciaand” and “Dysgonomonasintestinal”: spaces are missing here. The same problem occurs at other places, too. Please check the whole manuscript yourself.

Line 40-41: “whose larvae consume various organic wastes of environmental concern”. This is a strange formulation as it appears as if the larvae are looking for substrates with environmental concern. This is not the case. They can consume a wide range of substrates and this phenomenon can be applied in a positive way to feed them residues with environmental concern.

Line 52: Language problem. Replace “challenged larvae increased phenol oxidase” by “challenged larvae with increased phenol oxidase”.

Line 54: Replace “To combat infection, insect relies on innate defense reactions to defense pathogens” by “To combat infection, the insect relies on innate defense reactions to inhibit pathogens”.

Line 55: Replace “phenoloxidase” by “phenol oxidase”.

Line 57: “GNBP3 and PGRP-SA”: please clarify abbreviation.

Line 62 and following: the explanation of a Toll-like receptor as written here may not be clear to all readers. Please try to be more clear, for instance by replacing “Toll-like receptors (TLRs) are a class of type I membrane receptors with an extracellular amino terminus and a conserved cytoplasmic region. TLRs, which widely exist in antigen-presenting cells, are a pattern receptor involved in recognizing molecular structures (e.g., PAMPs) specific for microbial pathogens and have an important effect on innate and adaptive immune response.” by “Toll-like receptors (TLRs) are proteins present in the membranes of cells that are part of the immune system and recognize invaders (sentinel cells). They are a class of type I membrane receptors with an extracellular amino terminus and a conserved cytoplasmic region. TLRs recognize specific molecular structures in microbial pathogens and this recognition activates innate and adaptive immune responses.”

Line 66: “With routine microbial burdens, such as those found in the absence of infection, the Toll pathway at low activation levels.” A verb is missing in this sentence.

Line 75: Replace “Microbial flora modulate” by “The microbiota modulates”.

Line 78- 79: “to protect themselves against pathogenic microorganisms, such as bacteria and fungi”. And what about entomopathogenic viruses?

Line 79: “Kumar et al. (2010) [11] and Mitochondria in Anopheles gambiae Giles (Diptera: Culicidae) intestinal cells and Enterobacter of intestinal bacteria”: something is wrong with the construction of this sentence. I do not understand this sentence.

Line 88: “Recent surveys of black soldier fly gut microbiota revealed a diverse community dominated by Bacteroidetes and Proteobacteria[15].” Please include the following references here: https://doi.org/10.1128/AEM.01864-18, https://doi.org/10.1007/s00248-018-1286-x.

Line 102: replace “whether BSFL reduces suppression on zoonotic pathogens after BsfDuox-TLR3 RNA interference 104 (RNAi)” by whether suppression of zoonotic pathogens by BSFL is reduced after …”.

Line 110: Please include how you sterilized the feed (type of treatment and treatment parameters, such as time and temperature or radiation dose in the case of irradiation). If it was done by autoclavation, how did you make sure that nutrients for the larvae were not affected by the sterilization process? If no measurements were taken for this, can you guarantee that the growth of the larvae was normal after possible nutrient breakdown and that the model you study is representative for the large scale situation?

Line 111: what type of bran was use here? Wheat bran or from another cereal? Please make sure that your Materials and Methods are written in a way they can repeated by other researchers. For that reason, the type of bran should be specified.

Line 112: Please indicate the duration of each washing step so that other researchers can repeat the work.

Line 115: How was homogenisation performed? Equipment? Process parameters?

Line 120: “equal amounts of sterile distilled water”: equal to what? How did you determine this? Please specify.

Line 124: please mention supplier of the medium, because the medium composition can differ (slightly) from one supplier to another.

Line 138: Did you perform the appropriate confirmation tests on the presumptive colonies as is to be done for Baird-Parker medium? If not, why not and how sure are you that you counted Staphylococcus aureus colonies? Can you exclude that non-Staphylococcus aureus colonies, present in the background microbiota, have also grown on the plates? In other words, how specific was your counting method?

Lines 130 to 136 and 145 to 150 are the same. Please avoid to copy-paste the text and replace lines 145 to 150 by mentioning that sample processing to perform the counts was performed in the same way for Salmonella as for Staphylococcus.

Line 154: same remark as for Staphylococcus: did you perform confirmation tests, as is a standard rule in microbiology?

Line 139 and 155: you mention results were expressed in cfu/g but in the figure you mention log cfu/g. Therefore, mention log cfu/g in your text as well.

Line 167: “data not published” Is this still correct? Why not refer to https://doi.org/10.1038/s41422-019-0252-6?

Line 178: “Third-instar larvae (8 days old) were fed with an artificial diet containing concentrated microbe solution (1×108 CFU/ml),”: You mention the cell density of the inoculation suspension, but you do not mention how much of this solution was used per gram substrate. In this way, your method cannot be repeated by another researcher.

Line 184: “The resulting bacterial counts in each sample were adjusted to 1×108 CFU/ml by aseptic distilled water.” As I understand, this means that you prepared the cell suspension, you plated and incubated it and then counted the colonies. How long did this step take? If it took a few days (for incubation), how can you be sure that in the suspension all cells remained alive? Typically in microbiology, to assess the inoculum size, turbidity is measured and a McFarland standard is prepared.

Line 185: “The larvae fed with a 5% sucrose diet only served as the control.” Why did you use a sucrose as control? Can you exclude that the sucrose did not affect the growth and the intestinal microbiota of the larvae and hence that this was a good control? Please explain.

Line 188: “The microorganisms used in this study were pathogens S. aureus and Salmonella spp.” This sentence is redundant because you mentioned this many times earlier in the text. Please delete the sentence.

Line 191: Is a sample of three larvae representative for the whole batch? Please prove so.

Line 195: Language problem. Replace “The amplification program was consist of” by The amplification program consisted of”

Line 208: What do you mean with “rapidly”? How long did the dissection of one larvae take maximally?

Line 209: “The dissected intestines were ground with PBS.” Explain how. Can you assure that no ROS were lost during grinding?

Line 211: You mention that you measure the tissue and centrifuged it. Do you mean the ground tissue? If so, please add the word “ground”.

Line 207 says you measured TOTAL in vivo ROS, but in Line 209 you say you took the dissected intestines. Can you be sure that ROS only occur in the intestines of the larvae and thus that by taking the intestines you indeed determine all ROS? Can there be ROS in other tissues of the larvae as well and would the title in Line 207 better be “Measurement of intestinal ROS”?

Line 224-225: “primer pairs used in dsRNA synthesis are shown in Supplemental Table 1.1 μg PCR product was used”. Some words appear to be missing here: is 1.1 the number of the Table you refer to or is it the amount of PCR product used?

Line 281: “Each experiment was repeated three times”. Formulated this way, a reader may think that each experiment was performed a first time, and then later, AFTER the first experiment a second time, and again later a third time and finally, after the third time a fourth time. This is was is meant with REPEATED three times. However, this is not consistent with what you write in Line 161, where you say that the experiment was PERFORMED three times. Please make sure this is consistent in your text.

Line 284: “Significant was set at p < 0.05”; This is not a correct sentence. I guesse you mean “The significance level was set at p < 0.05.” (and ending the sentence with a point).

Line 300: “using the sterilized feed”: Why did you do this experiment in sterile feed? Please explain this choice in the paper. Did you expect that the attenuation would not work when the feed would not be sterilized? What is the possible impact of the absence of the background microbiota in the feed on the results?

Line 304: Replace “BSFL reduced the inhibition of S. aureus at all time points, and the interference effect was maintained” by “the inhibition by the larvae of s. aureus was reduced at all time points and the effect was maintained”, if this is what you mean.

Line 336: Replace “Infection of zoonotic pathogens” (which means that the pathogens get infected) by “Infection of zoonotic pathogens” (which means that the pathogens are infecting other organisms, here the larvae, I guess this is what you mean).

Line 366: “beneficial symbiont bacteria”: your data are based on the statement that Providencia and Dysgonomonas are symbionts. I have two questions here: (1) provide ample references from literature that these genera are indeed symbionts for BSFL to prove your statement and (2) can you assure you did not miss other symbionts? What if there are other symbionts in BSFL that are not discovered or described yet and that you do not take into account and that react completely different than the two you mention in your study?

Line 367: Replace “decreased by 3.4% compared with” by “were 3.4% lower than”.

Line 371: “the depletion of these two immune genes in BSFL reduced AMP and ROS production, leading to the decrease in symbiont bacteria and increase in pathogenic bacteria”: It is clear that the reduced expression of these genes reduced AMP and ROS and in this way resulted in an increase in pathogenic bacteria. However, it is not clear to me that the reduction of the symbionts is a direct result of the reduced gene expression, as you seem to indicate. Could it be the case that the gene expression was reduced, that therefore that pathogens were not reduced and that as a consequence of that latter fact the symbionts could not dominate due to competitive exclusion by the outnumbering pathogens? In other words, the reduced expression leads to an increase in pathogens and this in its turn leads to a reduction in symbionts, and not (as you write) a reduced expression leading to a reduction in symbionts and this in its turn to an increase in pathogens. Please clarify very well what is cause and what is result.

Line 382: replace “were actively” by “were active”.

Line 409: “At 4 days, Duox-TLR3 RNAi larvae with a decrease of 12% and 14.8%abundance of Providencia and Dysgonomonasthan control, respectively.” Where is the verb in this sentence?

Line 413: what do you mean with “minor bacterial”? Please rephrase in proper English.

Line 418: Replace “a minor bacterium” by “ a bacterium with low relative abundance”.

Line 432: Replace “microbiota” by “microbiota”.

Line 466: Replace “cell” by “cells”.

Line 743: “By contrast, Duox inactivation leads to uncontrolled ??? of bacteria in the gut”: it seems a word is missing here.

Line 484: Replace “domain” by “domains”.

Line 506: Replace “B.Dorsalis” by “B. dorsalis”.

Line 516: “BSF specifically belongs to the γ-proteobacteria”. This is confusing since BSF is not a bacterium. Please rephrase in a correct way.

Line 519: “Enterobacteriaceae may plays”. This sentence construction is not correct. Please reformulate.

Line 522: “Providencia spp. are gram-negative bacteria that was one of Enterobacteriaceae family”. Sentence construction is not correct. Please reformulate.

Line 525: “which could resistant to”. Same remark.

Line 525: “Meanwhile …” This sentence is too long and difficult to read because it only ends at line 530. Please reformulate.

Line 536: Replace “P. Aeruginosa” by “P. aeruginosa”.

Line 541: “By contrast, oriental BSF did not exhibit pathology associated with Pseudomonadaceae.” Where do you have this information from? Please refer to literature to support this claim.

Reviewer #2: In this manuscript Huang et al. analyze the role of Duox and Toll pathway in suppressing microbial pathogens in black soldier fly. By using different approaches the authors try to demonstrate that the two factors regulate pathogens by acting on the gut microbiota of the larva.

The mechanism used by the insect to discriminate resident bacteria from pathogen microorganisms is surely intriguing. Moreover the possibility to address this aspect in Hermetia illucens opens new applicative perspectives. However I think that the manuscript has many weak points:

a. The presentation of results is quite confused and it is difficult to follow the flow of the experiments. A language revision is necessary and the paper is full of typos.

b. The authors seem not to be aware of the current literature. Some examples:

- many papers on BSF gut microbiota have been recently published (De Smet et al., 2018; Bruno et al., 2019; Jiang et al., 2019), while this article cites only a study on bacteria associated to BSF (Zheng on et al., 2013)

- the authors do not mention a recent paper on BSF genome by the same research group (Zhan et al., 2019)

- previous studies on BSF immune response have been overlooked (Vogel et al., 2018; Cickova et al., 2015)

c. No information on the criteria for the choice of the bacterial concentrations administered to the larvae is reported.

d. It is not clear why they used third instar larvae since this is a stage where Duox gene is weakly expressed.

e. I’m concerned about the interpretation of some experiments. In fact the authors consider only some of the results to draw their conclusions, while others are completely overlooked. Some examples:

- Figure 7B: 4 and 8 days have an opposite trend

- Some data are discussed although not statistically significant

- Salmonella spp. and S. aureus differently induce the expression of Duox gene

f. The efficacy of RNAi in insects is strongly debated (see Scott et al., 2011; Terenius et al., 2011). This approach for gene silencing in Hermetia is new and there are no previous studies on this topic. Since most of the experiments of the current study are based on RNAi, further evidence is necessary to support qPCR data and demonstrate the efficacy of RNAi in this insect model.

g. Duox and TLR3 genes have been cloned starting from a comparison with Drosophila homologs, but Discussion on these two proteins considers B. dorsalis.

h. The authors should better argue why they focused attention on TLR3.

i. Discussion needs to be focused. In addition, some aspects are not relevant to the current study (see structural characterization of Duox and TLR).

I. Supplementary information needs to be revised. For example Figure S8 is blurry and it is not possible to read the text.

Reviewer #3: PONE-D-19-31548

Dual oxidase gene Duox and Toll-like receptor 3 gene TLR3 in the Toll pathway suppress zoonotic pathogens through regulating the intestinal bacterial community homeostasis in Hermetia illucens L.

Huang Y, et al.

Overall an interesting study. The manuscript has multiple small grammatical, spacing, tense and syntax errors, which make the manuscript difficult to read at points and require correction, some of which are listed below. The method section description is weak and requires further explanation and clarification. The statistical analyses in the results section require clarification as to how the analyses were done, how the data was grouped, as well why student T-test were done on time course data.

Materials and Methods

There is no validation of the sterilization method presented not a reference to a paper which validated the sterilization method. You assume that these larvae are sterilized, please present evidence for validation of this procedure. Please explain how larvae were sterilized, such as... how long were larvae washed, were they agitated or just left to float, etc. Please explain how BSFL were exposed to the pathogen, you only explain state how the control was exposed. Also please give more detail, such as amount of solution mixed in the manure, how much manure and how it was introduced? Was it poured on top, mixed in, and at what temperature did exposure occur, how were they contained/what were they exposed in, etc.? Were only 60 larvae exposed or did you take 60 larvae from a large group (this is unclear the way it is stated). What is meant by “approximately”, did you not count how many you dissected?

Please include website urls for references 20 and 21.

Line #

This manuscript has many spacing errors.

54 - …insects rely…to defend against…

77 - define Bd

80 – How does Kumer et al (as a reference) increase ROS? This sentence does not make sense.

85 – What is expression throughan? Again, poorly constructed sentence.

105 – …its relationship…

121 - ..replicated three times.

123 - A period is required “….detected[20].”

143 – distilled water that was then stirred…

143 – What is meant by “and used overnight”?

135, 150 – What is meant by “Pathogen counts were determined using a selective enrichment.”? Were the samples enriched first? If so, how were counts made?

154 - The mean Salmonella spp (CFU/g) of triplicate plates was determined.

159 - … by using TRIzol… according to manufacturer instructions.

160 – a full listing of the supply company used is required.

164; 194– a full and accurate listing of the supplies and company used is required.

195 - …amplification program consisted of…

205 – there are multiple primers listed in this reference please be specific which primers were used.

290 – what is meant by “in the natural environment” and “under normal pig manure conversion circumstances” this was not adequately explained in the methods.

Please use the same scale on your figures (1 and 2) so that they can directly be visually compared. Use a solid-no pattern for one of your data groups so they can be more easily distinguished form your pattern labels.

306 – The graphs do support a complete loss of inhibitory effect until after 2 days. They do support a reduction in the inhibitory effect seen with Fig 1.

324 – Be careful of your wording and punctuation, it appears you are saying the both Salmonella and Staph induced 4.05-times the Duos gene at 12 hours and this is not reflected in the graph.

325 - The same is true for this statement, make sure it is clear you are talking about Staph. The 24 hr increase is not shown on the graph? Put the significance designations directly above the error bars not at the top of the graph? Not sure students T-test is the proper analyses nor presentation by histogram. These were time courses, so you should be looking at the total time course differences via a line graph and an ANOVA (this would be true for all of your time course experiments presented in this manuscript. Also be clear in your statistical description what groups are being compared, in other words are you comparing only Staph to Staph at different time points or are you comparing all sample groups at the one time point to each other. This is not clear.

326 - I do not see on the graphs where Salmonella shows significant difference in Duox? Are you talking about 4 and 8 hours, if so where are the significance designations for Salmonella? Please clarify your statement.

340 – Analyses for quantifying effector gene dif is not described in the Methods section; nor is cecropin, ubiquitin and stomoxyn.

349 & 352 – An example that egfp RNAi and egfpRNAi is presented in these two different ways, with and without spacing. Be consistent throughout the manuscript. Also correct any capitalization of the species name (such as in Ln # 506, 536).

382 – active

Figures 8 and 9 – the labels and legends are too small, and the resolution of the figures is too low. These require better quality graphics. Dendograms should be shown to demonstrate the relatedness of the community composition.

409 – incomplete sentence.

420 – Where did you demonstrate that the relative abundance of Comamonas was directly tied to reduced ROS and AMP?

Fig 8 – Again keep axis values the same so direct comparisons can be made. Put significance markers on these graphs.

429 – Tie these last three sentences in better to the immune expression theme and the observations in your study.

461 – This is true only at Day 8 and with Chao and ACE. So this blanket statement is incorrect.

471 – fix spacing

473 - What does …uncontrolled of bacteria in the gut… mean?

516 - …belonging…

518 - …could potentially…

520 - …play…

522, 524, 525-528, 562 – poor sentence structure

531 – need a reference

389, 424, 438, 446, 498, 549 – avoid the first person “We”

Reviewer #4: • The following statement from the abstract is misleading: “Currently, it is the only insect approved for such purposes in Europe, Canada, and the USA”. A number of other species of insects can be legally used for feeding animals.

• Line 274: The R packages used should be cited in accordance with R software citation practices. These proper citation can be found in the software library.

• Line 316-319: why would these genes be less expressed at specific intermediary instars? What could be the explanation for that?

• Line 328-331: These results show that gene expression is increased. When the genes do express you refer to a plot where you show the increase in H202, but don’t discuss this in the text.

• Could the authors explain more clearly the mechanism by which the gene expression reduces the microbial load? Is it only H2O2, or was that the only measurable bioactive compound?

• Is there any insight into how activation of these genes reduces activation of other genes as a tradeoff, that might reduce growth performance in farming? In other words, what is the burden on the animal? Is this later described in line 438?

• Lines 349-375 show what as observed regarding time to recover expression of certain genes. Could the authors explain why the larvae needs the length of time observed to recover from silencing? What activities are underway that take the observed number of days to recover normal activity? Maybe this is explained in Lines 376-386, but it is not written in a way that is clear to me if this connection is being made.

• Line 432: a comparison is made to mosquitos. I find these types of comparisons relevant since they show how BSF act compared to other species. More clear comparisons with other species and how they react, especially quantitative ones, would be welcome. In particular, it would be very valuable to see how much H202 BSF produce relative to other members of Diptera under similar conditions. Do we see much higher output, which corresponds to their well-known robustness at the larval stage?

• Much is made of BSF’s ability to manage pathogens in industry. Could you please use part of the discussion to give direction to where more work needs to be done to confirm if the ability of BSF to reduce pathogens can be relied upon to produce safe raw larvae for the feed and food industries? Or can you explain how your results show that BSF will not be able to completely resolve a serious safety issue and that other steps will be needed, like heat treatment.

6. PLOS authors have the option to publish the peer review history of their article (what does this mean?). If published, this will include your full peer review and any attached files.

Reviewer #1: No

Reviewer #2: No

Reviewer #3: No

Reviewer #4: No

---

## [Author Response · Author response to Decision Letter 0]

11 Feb 2020

Dear Editor,

Journal of PLoS One

Kindly refer to your email sent on 2019/12/30 regarding the reviewers’ comments on our manuscript ID WM-19-31548 entitled "Dual oxidase Duox and Toll-like receptor 3 TLR3 in the Toll pathway suppress zoonotic pathogens through regulating the intestinal bacterial community homeostasis in Hermetia illucens L.". Based on his/her kind comments, we have carefully modified the manuscript. We submit the manuscript with track and no track version according to you requirements. The sequencing data of black soldier fly larvae gut bacteria 16S rDNA high-throughput sequencing has been uploaded. Serial number of original sequence data: bioprojectID PRJNA600829 and SRP247530 were added in Materials and Methods section of text. We have thoroughly revised as per suggestions of reviewers. The English language has been revised by a native English speaker scientist Professor Jeffery K. Tomberlin from Texas A &M University, USA .Thank you very much for the encouragement and providing very useful suggestions to improve the quality of our manuscript. We hope that the revised-manuscript now will meet the standards for publication in your esteemed journal.

---

## [Decision Letter · Decision Letter 1]

25 Feb 2020

PONE-D-19-31548R1

Dual Oxidase Duox and Toll-like receptor 3 TLR3 in the Toll pathway suppress zoonotic pathogens through regulating the intestinal bacterial community homeostasis in Hermetia illucens L.

PLOS ONE

Dear Dr. Zhang,

Thank you for submitting your revised manuscript to PLOS ONE.

Reviewer has major reservation about your work. I side with this reviewer and would give you one final opportunity to revise your paper further. Failure to satisfy the reviewer fully might result in rejection of your further revised paper.

We invite you to submit a revised version of the manuscript that addresses the points raised during the second round of review.

We would appreciate receiving your revised manuscript by Apr 10 2020 11:59PM. To enhance the reproducibility of your results, we recommend that if applicable you deposit your laboratory protocols in protocols.io, where a protocol can be assigned its own identifier (DOI) such that it can be cited independently in the future. For instructions see: http://journals.plos.org/plosone/s/submission-guidelines#loc-laboratory-protocols

We look forward to receiving your revised manuscript.

Kind regards,

Dong-Yan Jin

Academic Editor

PLOS ONE

Reviewers' comments:

Reviewer's Responses to Questions

**Comments to the Author**

1. If the authors have adequately addressed your comments raised in a previous round of review and you feel that this manuscript is now acceptable for publication, you may indicate that here to bypass the “Comments to the Author” section, enter your conflict of interest statement in the “Confidential to Editor” section, and submit your "Accept" recommendation.

Reviewer #2: (No Response)

2. Is the manuscript technically sound, and do the data support the conclusions?

Reviewer #2: Partly

3. Has the statistical analysis been performed appropriately and rigorously? 

Reviewer #2: No

4. Have the authors made all data underlying the findings in their manuscript fully available?

Reviewer #2: Yes

5. Is the manuscript presented in an intelligible fashion and written in standard English?

Reviewer #2: Yes

6. Review Comments to the Author

Reviewer #2: This is the revised version of a manuscript by Huang et al. in which the authors analyzed the role of Duox and Toll pathway in controlling microbial pathogens in Hermetia illucens.

Although the authors improved the manuscript and the text is now more fluent, they did not really address most of the questions raised during the first revision. Moreover some of their answers are cryptic. Some examples:

a. Bacterial concentration. Yao Z. et al., 2016 is a study on B. dorsalis, a different insect species. Why did the authors refer to this paper for their experiments on BSF? Did they perform preliminary assays in BSF to demonstrate that “low concentrations of bacteria cannot activate the immune system”, as stated in the rebuttal letter?

b. I still do not understand why third instar larvae were used (see answers as “four instar larvae were feathered”??? “fifth instar larvae were too old to take out the intestine”???)

c. Interpretation of some experiments. I do not see any information in the new text about the points that I indicated in the previous revision (see trend of some markers, statistics, expression of Duox gene, etc.)

d. RNAi. The authors just say that RNAi works in Drosophila, they declare to be able to use Crispr/cas9 gene editing in Hermetia and that they could not do any additional experiments since one of them left the laboratory. What does this mean? This does not reply to my question on RNAi. This is the first report on the use of RNAi in BSF, thus I think that this aspect is worthy of a deeper investigation.

e. Some of the revised points indicated in the cover letter are not present in the text (see for example L518 for B. dorsalis or L66 for TLR3)

f. Although Discussion has been revised, it still contains some information that, in my opinion, it not relevant to the current study (see for example structural characterization of Duox and TLR). Moreover, I’m still convinced that the whole section should be more focused.

g. Many typos are still present in the text.

7. PLOS authors have the option to publish the peer review history of their article (what does this mean?). If published, this will include your full peer review and any attached files.

Reviewer #2: No

---

## [Author Response · Author response to Decision Letter 1]

25 Mar 2020

Response to reviewer 2 comments

a.Bacterial concentration. Yao Z. et al., 2016 is a study on B. dorsalis, a different insect species. Why did the authors refer to this paper for their experiments on BSF? Did they perform preliminary assays in BSF to demonstrate that “low concentrations of bacteria cannot activate the immune system”, as stated in the rebuttal letter?

Response: Thank your good question. Because Hermetia illucens and B. dorsalis is close relatives and both belong to Diptera. Their BsfDuox and BsfTLR3 were similar. They did not perform preliminary assays in BSF to demonstrate that “low concentrations of bacteria cannot activate the immune system”. We just did some preliminary experiments to explore the suitable concentrations of bacteria for BSFL infect treatment.

b.I still do not understand why third instar larvae were used (see answers as “four instar larvae were feathered”??? “fifth instar larvae were too old to take out the intestine”???)

Response: The third instar is the right size to pull out the intestines. The other larvae are either too small or have no intestines to pull out.

c.Interpretation of some experiments. I do not see any information in the new text about the points that I indicated in the previous revision (see trend of some markers, statistics, expression of Duox gene, etc.

Response: Thanks for your suggestion, we have revised it in text(L130-133, L154-L155, L164-171, L183-L184, L190-L193, L199-L203, L212-L213, L224, L228-L229, L233-L236, L239-240, L245-L246, L247-L254, L270-L271, L283, L327-L328).

d.RNAi. The authors just say that RNAi works in Drosophila, they declare to be able to use Crispr/cas9 gene editing in Hermetia and that they could not do any additional experiments since one of them left the laboratory. What does this mean? This does not reply to my question on RNAi. This is the first report on the use of RNAi in BSF, thus I think that this aspect is worthy of a deeper investigation. 

Response: Our experiment is to study the black soldier flies, using crisper-cas system and RNAi technology to interfere the genes of BSF have achieved good results(Zhan et al 2019). RNAi technology worked well in Drosophila and Caenorhabditis elegans. Schistocerca americana et al. RNAi also has drawbacks and obscure points, However, such as those related to differences in species sensitivity,there are hundreds of research papers to prove the effectiveness of RNAi in insects. The QPCR test could verify the RNAi results, each QPCR experiment was repeated three times.

e:Some of the revised points indicated in the cover letter are not present in the text (see for example L518 for B. dorsalis or L66 for TLR3)

Response: Thank you for your suggestion. In L518 I wrote Enterobacteriaceae bacteria instead of a B. dorsalis ,which appears in the cover letter. In L66 I wrote TLR3,which also appears in the cover letter.

f:Although Discussion has been revised, it still contains some information that, in my opinion, it not relevant to the current study (see for example structural characterization of Duox and TLR). Moreover, I’m still convinced that the whole section should be more focused.

Reponse: Thank you for your suggestion. We have deleted not relevant contents in discussion section such as structural characterization of Duox and TLR and other not relevant information(L549，L599-L603，L621-L627，L643).

g: Many typos are still present in the text.

Reponse: Thanks for your suggestion, We have double checked and revise it in whole text(L535，L539 ,L663-L888 et al.).

---

## [Editor Report · Decision Letter 2]

27 Mar 2020

Dual Oxidase Duox and Toll-like receptor 3 TLR3 in the Toll pathway suppress zoonotic pathogens through regulating the intestinal bacterial community homeostasis in Hermetia illucens L.

PONE-D-19-31548R2

Dear Dr. Zhang,

We are pleased to inform you that your manuscript has been judged scientifically suitable for publication and will be formally accepted for publication once it complies with all outstanding technical requirements.

With kind regards,

Dong-Yan Jin

Academic Editor

PLOS ONE
---

## [Editor Report · Acceptance letter]

3 Apr 2020

PONE-D-19-31548R2 

Dual Oxidase *Duox* and Toll-like receptor 3 *TLR3* in the Toll pathway suppress zoonotic pathogens through regulating the intestinal bacterial community homeostasis in *Hermetia illucens* L. 

Dear Dr. Zhang:

I am pleased to inform you that your manuscript has been deemed suitable for publication in PLOS ONE. Congratulations! Your manuscript is now with our production department. 

With kind regards,

on behalf of

Professor Dong-Yan Jin 

Academic Editor

PLOS ONE